# Allosteric competition and inhibition in AMPA receptors

W. Dylan Hale[1,2], Alejandra Montaño Romero[1,2,8], Cuauhtemoc U. Gonzalez [3,4,8], Vasanthi Jayaraman[3], Albert Y. Lau [2]✉, Richard L. Huganir [1,5]✉ & Edward C. Twomey [1,2,6,7]✉

Excitatory neurotransmission is principally mediated by α-amino-3-hydroxy-5-methyl-4-isoxazolepropionic acid (AMPA)-subtype ionotropic glutamate receptors (AMPARs). Negative allosteric modulators are therapeutic candidates that inhibit AMPAR activation and can compete with positive modulators to control AMPAR function through unresolved mechanisms. Here we show that allosteric inhibition pushes AMPARs into a distinct state that prevents both activation and positive allosteric modulation. We used cryo-electron microscopy to capture AMPARs bound to glutamate, while a negative allosteric modulator, GYKI-52466, and positive allosteric modulator, cyclothiazide, compete for control of the AMPARs. GYKI-52466 binds in the ion channel collar and inhibits AMPARs by decoupling the ligand-binding domains from the ion channel. The rearrangement of the ligand-binding domains ruptures the cyclothiazide site, preventing positive modulation. Our data provide a framework for understanding allostery of AMPARs and for rational design of therapeutics targeting AMPARs in neurological diseases.

Glutamate (Glu) is the principal neurotransmitter in the brain. Neurons in the brain use Glu at excitatory synapses, where Glu is released by a presynaptic neuron and received by a postsynaptic neuron[1]. Ionotropic Glu receptors (iGluRs) in the membrane of the postsynaptic neuron bind Glu and allow cations to enter, depolarizing the postsynaptic membrane[2]. Specialized iGluRs, α-amino-3-hydroxy-5-methyl-4-isoxazolepropionic acid receptors (AMPARs), initiate the depolarization of the postsynaptic neuron and contribute to the activation of other iGluR subtypes[3].

Dysregulation of AMPARs contributes to neurological disorders including schizophrenia, anxiety, chronic pain, epilepsy, learning impairment, Alzheimer disease and Parkinson disease[2]. AMPAR allosteric modulators are a promising avenue for therapeutics as they

allow AMPAR function to be positively or negatively tuned independent of Glu binding. However, despite the central role of AMPARs in synaptic signaling and their roles in human diseases, only a single molecule, perampanel (Fycompa), is approved by the US Food and Drug Administration (FDA) for targeting AMPARs for therapeutic benefit[2,4]. Perampanel is approved for treatment of epilepsy[5] and perampanel-like molecules (PPLMs) show promise in treating broad neurological disorders.

PPLMs are noncompetitive AMPAR inhibitors typified by the prototype compound 4-(8-methyl-9H-1,3-dioxolo[4,5-h][2,3]benzodiazepin-5-yl)-benzenamine dihydrochloride (GYKI-52466)[2,6,7], which binds to the AMPAR transmembrane domain (TMD)[8]. PPLMs bind to the same site in the TMD and inhibit AMPAR channel function

[1]Solomon H. Snyder Department of Neuroscience, Johns Hopkins University School of Medicine, Baltimore, MD, USA. [2]Department of Biophysics and Biophysical Chemistry, Johns Hopkins University School of Medicine, Baltimore, MD, USA. [3]Center for Membrane Biology, Department of Biochemistry and Molecular Biology, The University of Texas Health Science Center at Houston, Houston, TX, USA. [4]The University of Texas MD Anderson Cancer Center UTHealth Houston Graduate School of Biomedical Sciences, The University of Texas Health Science Center at Houston, Houston, TX, USA. [5]Kavli Neuroscience Discovery Institute, Johns Hopkins University School of Medicine, Baltimore, MD, USA. [6]The Beckman Center for Cryo-EM at Johns Hopkins, Johns Hopkins University School of Medicine, Baltimore, MD, USA. [7]Diana Helis Henry Medical Research Foundation, New Orleans, LA, USA. [8]These authors contributed equally: Alejandra Montaño Romero, Cuauhtemoc U. Gonzalez. ✉e-mail: alau@jhmi.edu; rhuganir@jhmi.edu; Twomey@jhmi.edu

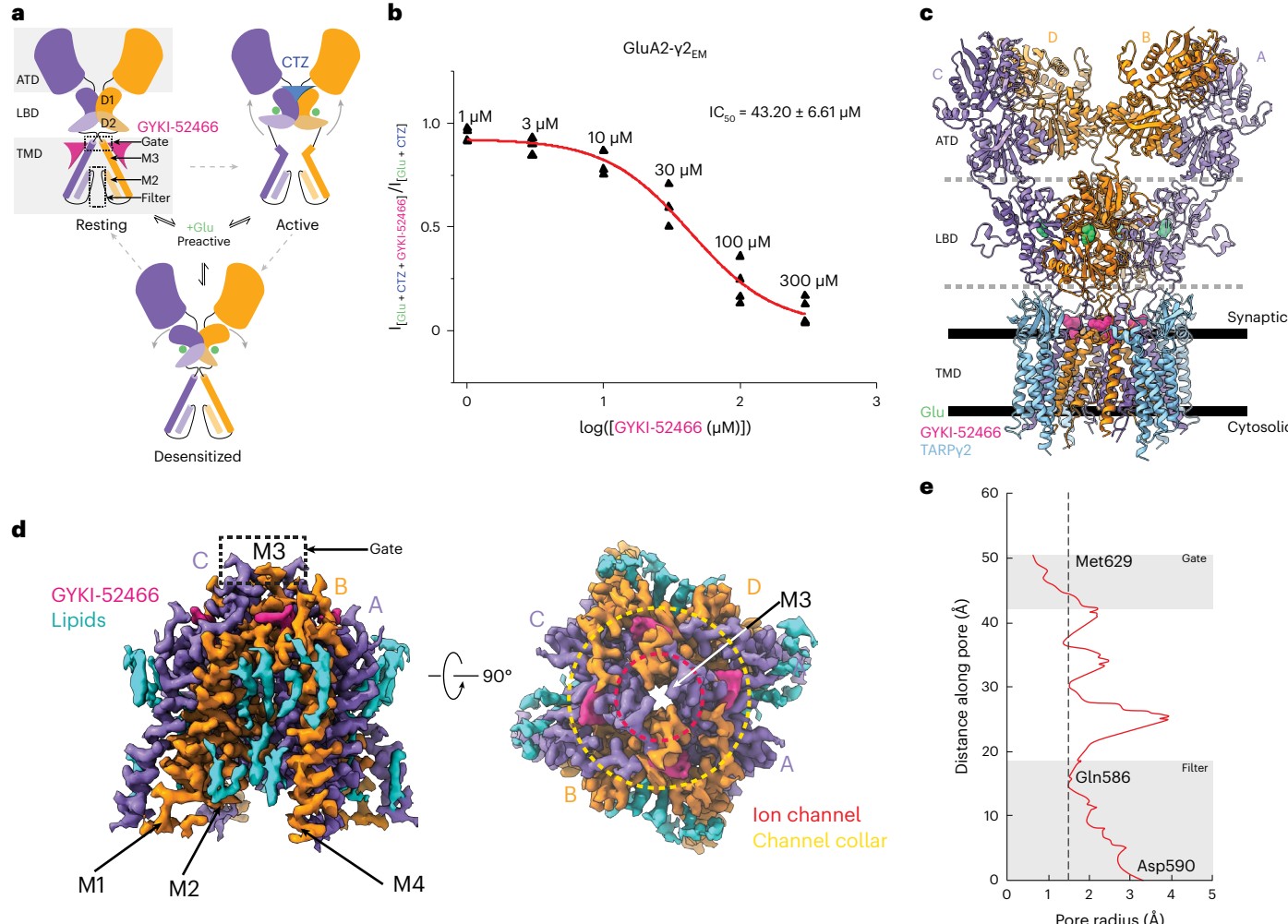

**Fig. 1 | Structure of the AMPAR allosterically inhibited state. a**, Schematic representation of the AMPAR gating cycle. Only two of four subunits are shown for illustration purposes. **b**, Concentration-dependent inhibition by GYKI-52466 of GluA2-γ2_EM residual currents in the presence of 1 mM Glu and 100 μM CTZ using nonlinear curve fit approach with the Levenberg–Marquardt iteration algorithm. For each concentration, data were obtained from at least three different cells. $IC_{50} = 43.20 \pm 6.61$ μM; $P = 0.00022$. **c**, Ribbon illustration detailing the structure of the AMPAR inhibited state, GluA2-γ2_IS-1. GluA2 subunits are purple (A and C) or orange (B and D) depending on their positions. GYKI-52466 (pink) is bound at all four TMD collar regions and each LBD clamshell is closed around Glu (green). TARPγ2 subunits (light blue) occupy all four auxiliary sites around the receptor. **d**, High-resolution details of the focused GluA2 TMD from cryo-EM reconstruction. Left: side view of the GluA2 TMD showing the M3 bundle crossing in a closed conformation. Right: top view showing the bundle crossing constricting access to the ion channel (red, dashed) and the relative location of the channel collar (yellow, dashed) with GYKI-52466 bound to all four GluA2 subunits. Lipids (blue) adorn the AMPAR TMD. **e**, Plot of the ion channel radius along the pore axis showing a constriction at the M3 bundle crossing gate. The dashed line represents the radius of a water molecule.

irrespective of channel state or membrane voltage[4,7–10]. PPLMs are effective at reducing epileptic behavior in mice and in vitro[11,12], as well as in human patients, and perampanel was recently used to reduce seizure burden in patients with rare mutations in synaptic genes including *GRIA2* (ref. 13) and *SYNGAP1* (refs. 11,14,15). However, treatment can produce side effects such as dizziness, somnolence and ataxia[16], underscoring the need for refined AMPAR inhibitors for treating neurological disorders. While the binding sites of PPLMs have been generally described[8], the precise mechanism by which PPLMs inhibit AMPAR function is unresolved. This is a major roadblock in therapeutically targeting AMPARs with improved inhibitors.

AMPARs are tetrameric ligand-gated ion channels, made up of GluA1–GluA4 subunits[2,17]. AMPARs couple extracellular binding of Glu to ion flux across the postsynaptic membrane through their ligand-binding domains (LBDs) that are directly coupled to transmembrane (TM) helices that form the cation channel[18,19]. Glu binding to the AMPAR LBDs initiates the gating cycle in which the receptors

transition through their main functional states: resting, activated and desensitized[2,20] (Fig. 1a). Linkers between the LBD and TMD enable the LBDs to control the state of the ion channel. The gating cycle is generally accommodated by a 'preactive' transition state that is short lived[20–25] (Fig. 1a). In this transition state, LBD clamshells are intermediately closed around Glu and the gating rearrangements associated with activation or desensitization are yet to occur. Thus, the transition state is a bifurcation point. Activation follows with full clamshell closure around Glu, where the lower half of the LBD clamshell (D2) moves closer to the upper half (D1) of the LBD[18,19] (Fig. 1a). Because AMPAR LBDs locally dimerize within the tetrameric receptor, coordinated clamshell closure maximizes the interface between the upper D1 lobes of LBD dimer pairs and increases separation of the D2 LBD lobes. Separation of the D2 lobes pulls apart the M3 helices that form the top of the channel gate, enabling cation influx through the upper vestibule, constituting the M3 helices, the M2 helices and a selectivity filter between M2 and M3 (Fig. 1a).

**Table 1 | Individual cell mean residual current, s.d. and number of sweeps for each GYKI-54266 concentration, along with the initial response in the absence of GYKI-52466**

| [GYKI-52466] (µM) | Mean residual current from Glu+CTZ+GYKI-52466 (pA) | s.d. (pA) | Number of sweeps | Initial mean residual current from Glu+CTZ (pA) | s.d. (pA) | Number of sweeps |
|---|---|---|---|---|---|---|
| 1 | −3,133.63 | 39.85 | 3 | −3,209.37 | 21.95 | 2 |
| | −169.22 | 5.39 | 4 | −185.05 | 4.09 | 6 |
| | −1,561.56 | 19.94 | 3 | −1,619.52 | 17.08 | 4 |
| | −240.30 | 7.91 | 6 | −261.97 | 5.12 | 4 |
| 3 | −31.87 | 1.47 | 4 | −37.72 | 1.31 | 6 |
| | −8,108.75 | 104.25 | 3 | −9,585.67 | 49.35 | 2 |
| | −229.67 | 5.10 | 4 | −247.38 | 5.30 | 5 |
| | −152.64 | 6.44 | 5 | −169.66 | 5.52 | 4 |
| 10 | −2,778.95 | 65.47 | 2 | −3,209.37 | 21.95 | 2 |
| | −143.85 | 6.19 | 3 | −185.05 | 4.09 | 6 |
| | −1,790.07 | 60.42 | 3 | −2,371.98 | 40.00 | 5 |
| | −230.18 | 6.20 | 4 | −265.72 | 7.21 | 3 |
| 30 | −18.86 | 2.09 | 5 | −37.72 | 1.31 | 6 |
| | −5,705.77 | 158.82 | 3 | −9,585.67 | 49.35 | 2 |
| | −174.54 | 7.78 | 3 | −247.38 | 5.30 | 5 |
| | −100.41 | 4.54 | 4 | −169.66 | 5.52 | 4 |
| 100 | −798.20 | 39.99 | 3 | −3,209.37 | 21.95 | 2 |
| | −66.33 | 4.17 | 3 | −185.05 | 4.09 | 6 |
| | −390.00 | 25.57 | 3 | −2,371.98 | 40.00 | 5 |
| | −34.70 | 3.65 | 4 | −265.72 | 7.21 | 3 |
| 300 | −1.41 | 0.60 | 4 | −37.72 | 1.31 | 6 |
| | −385.81 | 14.45 | 4 | −9,585.67 | 49.35 | 2 |
| | −41.93 | 3.72 | 2 | −247.38 | 5.30 | 5 |
| | −7.56 | 1.31 | 5 | −169.66 | 5.52 | 4 |
| | −16.67 | 1.97 | 10 | −131.32 | 2.01 | 6 |

Peak amplitudes were obtained in the presence of 1 mM Glu and 100 µM CTZ before the application of GYKI-52466 at different concentrations.

Desensitization occurs when LBD clamshells are maximally closed around Glu; however, instead of using this energy to pull apart the M3 ion channel gate, LBD dimer pairs roll away from each other, which minimizes the separation between D2 interfaces and reduces the tension applied to the M3 helices by the LBD–TMD linkers[21,24,26–30] (Fig. 1a). This keeps the channel in a closed state and protects the cell from excitotoxic influx. While there is an amino terminal domain (ATD), the major role of the ATD is in trafficking and assembly[31–36]; thus, we focus on the AMPAR LBD and TMD in this paper.

Allosteric modulators bind to AMPARs at sites distinct from the Glu-binding site and bias AMPAR function. Positive allosteric modulators such as cyclothiazide (CTZ) bind between the D1 lobes of local LBD dimers and enhance D1–D1 contact during activation, thus favoring activation and preventing AMPAR desensitization[21,37–39]. How negative allosteric modulators such as PPLMs prevent AMPAR activation is less clear. Mutagenesis and electrophysiology studies predicted that PPLMs act at an intersubunit interface between the LBD and TMD and prevent active-state transitions[40]. Subsequent studies of resting-state AMPARs bound to PPLMs identified a binding pocket within the TMDs of individual receptor subunits that make intersubunit contacts within the TMD[8,41]. Taken together, these studies suggest that PPLMs bind to the region of the TMD that is extracellular facing and prevent AMPARs from transitioning to the active state[2,8,40]. Several mechanisms have been proposed, including a wedge-like mechanism in which PPLMs prevent channel opening[8] or a mechanism that involves cross-linking adjacent GluA subunits within the TMD[10], preventing pore widening observed in

the active state[18]. These mechanisms share the common feature of disrupted transduction between Glu binding in the LBD and channel opening in the TMD but how this occurs is unknown because AMPARs have not been studied structurally in the presence of both Glu and PPLMs[8,41].

Pioneering studies on the mechanisms of PPLMs pointed toward an inhibition mechanism that competes with the positive allosteric effect of CTZ[8–10,21,40,42–46]. However, because CTZ modulates AMPARs by binding in the LBD and PPLMs bind in the TMD, how this competition occurs is unclear. We, therefore, hypothesized that, to compete against CTZ, which prevents desensitization, PPLMs must achieve inhibition by destabilizing the D1–D1 dimer interface between agonist-bound LBDs and promoting a conformational state that decouples Glu binding from channel opening. In this conformation, the D1 interfaces between LBD dimers would be separated, rupturing the CTZ-binding site and, thus, outcompeting CTZ for allosteric control of the AMPAR, as was originally proposed[40]. This mechanism would explain how positive modulators such as CTZ and negative modulators such as PPLMs compete to control AMPAR function despite binding at disparate sites. Such a mechanism has not yet been directly observed in AMPARs or any family of ligand-gated ion channels.

To test these ideas, we activated AMPARs in the presence of both GYKI-52466 and CTZ. Through cryo-electron microscopy (cryo-EM), single-molecule fluorescence resonance energy transfer (smFRET), electrophysiology and molecular dynamics (MD) simulations, we demonstrate that GYKI-52466 binding in the TMD decouples Glu binding from the ion channel by allosterically rearranging the AMPAR LBD into

**Table 2 | Cryo-EM data collection, refinement and validation statistics**

| | GluA2-γ2$_{IS-1}$ (EMD-43275),(PDB 8VJ6) | GluA2-γ2$_{IS-2}$ (EMD-43276),(PDB 8VJ7) |
|---|---|---|
| **Data collection and processing** | | |
| Magnification | ×130,000 | ×130,000 |
| Voltage (kV) | 300 | 300 |
| Electron exposure (e⁻ per Å²) | 40 | 40 |
| Defocus range (μm) | −1.0 to 2.6 | −1.0 to 2.6 |
| Pixel size (Å) | 0.93 | 0.93 |
| Symmetry imposed | C2 | C2 |
| Initial particle images (no.) | 1,258,087 | 1,031,751 |
| Final particle images (no.) | 123,729 | 130,474 |
| Map resolution (Å) FSC=0.143 | 3.50 | 4.85 |
| Map resolution range (Å) | 2–13 | 2.5–13 |
| **Refinement** | | |
| Initial model used | PDB 5WEO | GluA2-γ2$_{IS-1}$ |
| Model resolution (Å) FSC=0.143 | 4.2 | 3.50 |
| Model resolution range (Å) | 3.4–4.1 | 3.2–4.1 |
| Map sharpening B factor (Å²) | −65 | −120 |
| Model composition | | |
| Non-hydrogen atoms | 25,180 | 25,179 |
| Protein residues | 3,186 | 3,186 |
| Ligands | 4 | 4 |
| B factors (Å²) | | |
| Protein | 0.00/98.47/54.29 | 0.00/391.17/141.52 |
| Ligand | 0.00/23.22/7.37 | 0.01/9.83/4.80 |
| R.m.s.d. | | |
| Bond lengths (Å) | 0.006 | 0.004 |
| Bond angles (°) | 0.631 | 0.650 |
| **Validation** | | |
| MolProbity score | 1.61 | 1.64 |
| Clashscore | 6.38 | 6.96 |
| Poor rotamers (%) | 0 | 0.48 |
| Ramachandran plot | | |
| Favored (%) | 96.18 | 96.21 |
| Allowed (%) | 3.63 | 3.54 |
| Disallowed (%) | 0.19 | 0.25 |

an allosterically inhibited state. LBD rearrangements during inhibition prevent positive allosteric modulation by CTZ in the LBD by disrupting the CTZ-binding site. Our findings provide insights into how allosteric modulation is coordinated across AMPARs, demonstrate the mechanistic basis of allosteric competition between modulators and will invigorate structure-based drug design targeting AMPARs.

## Results

### Cryo-EM of allosterically inhibited AMPAR complexes

Previously, a fusion construct between the AMPAR subunit GluA2$_{flip}$ (edited to Gln at the Gln/Arg site) and the TM AMPAR regulatory protein (TARP)γ2, which enhances AMPAR activation, was used to solve the structures of AMPAR complexes and elucidate AMPAR gating mechanisms with cryo-EM[18,26,47–49]. We used the same fusion construct, GluA2-γ2$_{EM}$, in this study (Methods). The gating function of this exact construct and its modulation by positive and negative allosteric modulators were extensively validated previously[8,18,22,26,47–49].

To confirm inhibition by PPLMs and competition between PPLMs and CTZ in GluA2-γ2$_{EM}$, we used patch-clamp electrophysiology in HEK293T cells expressing GluA2-γ2$_{EM}$ (Extended Data Fig. 1a,b and Methods). We observed that GluA2-γ2$_{EM}$ currents rapidly desensitize when treated with 1 mM Glu and desensitization was ablated with 100 μM CTZ (Extended Data Fig. 1a), as expected[9,18,44]. In the presence of both 100 μM GYKI-52466 and 100 μM CTZ, the GluA2-γ2$_{EM}$ peak current following 1 mM Glu application was strongly reduced compared to CTZ alone (Extended Data Fig. 1a). This agrees well with previous electrophysiology studies on PPLM and CTZ competition in AMPARs in the absence of TARPs[9,10,40,42–46,50]. Thus, GYKI-52466 and CTZ both allosterically modulate GluA2-γ2$_{EM}$ and compete for influence over GluA2-γ2$_{EM}$ gating.

We generated concentration–response curves to fully characterize the competition between GYKI-52466 and CTZ in the GluA2-γ2$_{EM}$ construct (Fig. 1b, Extended Data Fig. 1b, Table 1 and Methods). GYKI-52466 inhibits GluA2-γ2$_{EM}$-mediated currents even in the presence of excess CTZ (Fig. 1b). We determined the half-maximal inhibitory concentration (IC$_{50}$) of GYKI-52466 in the presence of CTZ to be 43.20 ± 6.61 μM (P = 0.00022). This is a ~10-fold reduction in the IC$_{50}$ compared to GYKI-52466 alone on AMPAR–TARP complexes[51], which aligns well with the observed 10-fold reduction in GYKI-52466 IC$_{50}$ on AMPARs in the presence of CTZ[40,42,50].

We probed the precise mechanisms of allosteric competition with GluA2-γ2$_{EM}$. To achieve this, we purified GluA2-γ2$_{EM}$ from Expi293 Gnti⁻ cells (Extended Data Fig. 1c,d and Methods) and preincubated the receptors with CTZ. We activated these AMPAR complexes in the presence of GYKI-52466 to capture inhibited states through two different schemes (Extended Data Fig. 1e and Methods). In the first scheme (inhibited state 1, GluA2-γ2$_{IS-1}$), we mixed the CTZ-bound receptors with Glu and GYKI-52466 immediately before freezing. In the second scheme (GluA2-γ2$_{IS-2}$), the receptors were preincubated with GYKI-52466 in addition to CTZ and Glu was added immediately before freezing. Each approach resulted in similar inhibited states, with each domain in the structures only varying by a root-mean-square deviation (r.m.s.d.) of 0.3–0.4 Å (Extended Data Fig. 2a).

We focus our analysis on GluA2-γ2$_{IS-1}$ because of the higher data quality (Extended Data Figs. 3 and 4 and Table 2). The overall structures of the AMPAR complexes reveal key details of an inhibited AMPAR (Fig. 1c). There is an overall 'Y' arrangement of the receptor, with the two-layered extracellular domain comprising the ATD and LBD. The overall structure of the receptor shares similar topologies to previously determined structures from the GluA2-γ2$_{EM}$ construct, as well as purified AMPAR complexes from a native source[18,26,47,48,52–54]. All four GluA2 LBDs are Glu bound and immediately below the LBDs is the GluA2 TMD, which is fully occupied with four TARPγ2 auxiliary subunits. Four GYKI-52466 molecules are bound to the TMD along its extracellular-facing surface.

Cryo-EM reconstruction of the AMPAR TMD to 2.6 Å enables elucidation of key features of the AMPAR TMD during inhibition. The four GYKI-52466 molecules are wedged between helices at the top of the TMD (Fig. 1d). Importantly, the GYKI-52466-binding sites are adjacent to the ion channel in the channel collar region, similar to other PPLMs[8]. The collar channel forms a ring of solvent-accessible pockets for PPLMs that surrounds the M3 gate at the top of the ion channel (Fig. 1d). Lipids adorn the AMPAR TMD on both the extracellular-facing and the cytosolic-facing portions of the TMD (Fig. 1d) and are critical to plug cavities within the bilayer that would otherwise perturb the solvent accessibility of the ion channel (Extended Data Fig. 5). These

lipid sites are like those occupied in other cryo-EM studies of AMPARs, which suggests that these sites are critical for the structural integrity of the AMPAR TMD[27,55–57]. Next, we measured the ion channel radius, which indicates a closed channel; the upper channel gate, defined by Met629 at the M3 helix crossing, completely restricts channel access (<1.0 Å radius) to both water molecules and sodium ions (Fig. 1e). Both inhibited states captured in this study are markedly different from the resting-state AMPARs bound to PPLMs (Extended Data Fig. 2b).

While both the activator (Glu) and the negative allosteric modulator (GYKI-52466) are bound to the AMPAR (Extended Data Fig. 6), the positive allosteric modulator (CTZ) is absent from both cryo-EM reconstructions. This indicates that the states we captured are markedly different from previously captured states of AMPARs bound to PPLMs, as CTZ binds to both the resting and the activated states of the receptor[40,49,58]. Thus, GYKI-52466, at a binding site completely distinct from that of CTZ in the AMPAR LBD, allosterically outcompetes CTZ to control GluA2-γ2$_{EM}$.

A structural comparison of GluA2-γ2$_{IS-1}$, GluA2-γ2$_{IS-2}$ and PPLMs bound to resting-state AMPARs reveals that we captured a distinct, allosterically inhibited AMPAR conformation. While there are no notable differences between the GluA2 ATDs (r.m.s.d. = 0.7–1.0 Å; Extended Data Fig. 2b), there are major overall differences between the structures (r.m.s.d. = 6.1–7.0 Å), where the key differences among GluA2-γ2$_{IS-1}$, GluA2-γ2$_{IS-2}$ and resting-state AMPARs bound to PPLMs occur within the GluA2 LBD (Extended Data Fig. 2b). Thus, we posited that the major impact of allosteric inhibition by GYKI-52466 in the TMD is rearrangement of the LBD and we focused on the AMPAR LBD and TMD to discern the inhibition and competition mechanisms.

### The GYKI-52466-binding site

Reconstruction of the AMPAR TMD enabled precise building of the AMPAR TMD (Extended Data Fig. 6a). While previous studies solved the structure of other PPLMs in complex with resting-state AMPARs[8,41], GYKI-52466 binding in AMPARs remained structurally unresolved. To resolve GYKI-52466 binding, we used symmetry expansion on the AMPAR TMD from GluA2-γ2$_{IS-1}$ to reconstruct the binding site to 2.2 Å resolution (Extended Data Fig. 6b–e). This enabled us to characterize the complete binding pocket (Fig. 2a,b). GYKI-52466 is partially stabilized in the collar through a π-bond stack where GYKI-52466 is sandwiched between Phe623 at the top of the M3 helix and Pro520 on the pre-M1 helix (Fig. 2a). This differs only slightly from the binding pocket previously published for the structurally related compound GYKI-53655, GYKI-Br, and the structurally unrelated CP-465022, in which Phe623 is rotated away from the binding pocket[8] (r.m.s.d. = 1.5–1.6 Å; Extended Data Fig. 6f). Notably, Phe623 has a direct role coordinating perampanel in the channel collar in resting-state AMPARs[8,41] (Extended Data Fig. 6f), with an overall similar binding pocket (r.m.s.d. = 1.8 Å; Extended Data Fig. 6f).

The arrangement of Phe623 around GYKI-52466 that we observe may be attributable to the binding of Glu in the LBD driving a subtle conformational change that locks GYKI-52466 into the binding pocket during allosteric inhibition[40]. Van der Waals forces from five nearby residues, Ser516, Asn619, Ser615, Tyr616 and Asn791, also contribute to the binding site (Fig. 2b). Asn3 of GYKI-52466 is sandwiched between Tyr616 on M3 and Ser615 on M3 of an adjacent subunit. Therefore, GYKI-54266 is wedged between two AMPAR subunits in the TMD, similarly to the PPLMs[8] (Fig. 2b). While the GYKI-52466 pocket shares the same overall conformation as that reported for other PPLMs (r.m.s.d. = 1.5–1.8 Å), GYKI-52466 makes fewer contacts with pocket residues because of its smaller size and simpler structure. This may explain its relatively weaker affinity for AMPARs compared to other PPLMs[8–10].

During AMPAR activation, subunits in the B and D positions undergo the most dramatic conformational changes in the TMD to drive opening[18,28,56]. Kinking in the B and D M3 helices during activation directly impacts the PPLM-binding pocket[10,24] and we expected

the binding pocket around GYKI-52466 to be more compact in the B and D positions during inhibition. To assess this, we measured the distances between Pro520, Asn791, Ser615 and Phe623 (Fig. 2c). To our surprise, the binding pockets in each subunit were remarkably similar (Fig. 2d). On average, there was a ~12 Å distance between pairs Pro520–Asn791 and Ser615–Phe625, ~9 Å distance between Asn791–Ser615 and ~8 Å distance between Phe623–Pro520. Thus, the shape around the GYKI-52466-binding site is roughly the same at each subunit position, with an average solvent-accessible surface area of ~493 Å$^2$ around GYKI-52466. Thus, there are no discernible differences between subunit positions in the TMD in the inhibited state.

### GYKI-52466 decouples ligand binding from ion channel opening

To elucidate the inhibition mechanism, we compared our structure in the inhibited state to an activated AMPAR (Fig. 3a). The majority of the TMD was similar between the two states (r.m.s.d. = 1.0 Å; Extended Data Fig. 2c), except at the channel gate, which is formed by the top of the M3 helices (Fig. 3a). During activation, the M3 helices kink outward from the pore axis to open the channel[18,49,56,59]. This key movement is blocked by the presence of GYKI-52466 in the B and D AMPAR subunit positions because of the presence of GYKI-52466 in the channel collar (Fig. 3a, inset)[10]. However, there are no key differences between the GYKI-52466 B and D subunit and A and C subunit positions of the channel collar in the inhibited state (Fig. 2d). In addition, each individual LBD in the inhibited state is Glu bound, with a similar overall conformation to individual LBDs in the activated state (r.m.s.d. = 0.87 Å; Fig. 3b).

The inhibited LBD layer is markedly different from that in the activated state (Fig. 3a). While individual LBDs in each protomer share the same Glu-bound conformation (Fig. 3b), LBD dimers undergo a substantial conformational change to accommodate AMPAR inhibition. To assess these changes, we measured the distances between the D1–D1 and D2–D2 lobes in LBD local dimers, which are major indicators of the functional state of the AMPAR[24]. For example, during activation, the distances between D1 lobes in LBD local dimers are decreased as the D2 lobes separate to pull open the ion channel (Fig. 1a). During desensitization, the opposite occurs, where the D1 lobes separate and the D2 interface is minimized, which decouples Glu binding from the channel, allowing it to close (Fig. 1a).

In GluA2-γ2$_{IS-1}$, we measured the distances between the Cα atoms of Ser741 (D1 separation) and Ser635 (D2 separation) (Fig. 3c). The D1 interface is markedly separated (27 Å) compared to the D2 interface (16 Å). We then assessed how these separations fit with the conformational landscape of existing AMPARs (Fig. 3d). Generally, structures with a ≥26 Å distance between Ser741 residues in D1 lobes represent a desensitized state, while structures with a ≥27 Å distance between Ser635 residues in D2 lobes represent an active state, with resting-state structures representing a medium between the two separations. The activated state of AMPAR is exemplified by Protein Data Bank (PDB) 5WEO, while the resting state is exemplified by PDB 3KG2 and the desensitized state is exemplified by PDB 5VHZ (all PDB structures are mapped in Extended Data Fig. 7). The substantial rupturing of the D1 interfaces in both GluA2-γ2$_{IS-1}$ and GluA2-γ2$_{IS-2}$ places these LBD dimers squarely into the desensitized classification of LBD dimers. Critically, existing PPLM-bound structures in the PDB represent the resting state of the receptor because they are not Glu bound (Fig. 3d). This is marked by notable differences across the receptors between the PPLM-bound apo states and the inhibited states from this study (Extended Data Fig. 2b).

While the LBDs in local dimers are in a desensitized-like state, the total motion of the LBD layer reveals that allosteric inhibition is unique from desensitization. During desensitization, the A and C subunits roll away from their B and D partners to separate local dimers and decouple Glu binding from the ion channel[26,27] (Fig. 3e). In inhibition, we observe the opposite, with the B and D LBDs rotating 21° counterclockwise away from their A and C counterparts, which appear to maintain the position

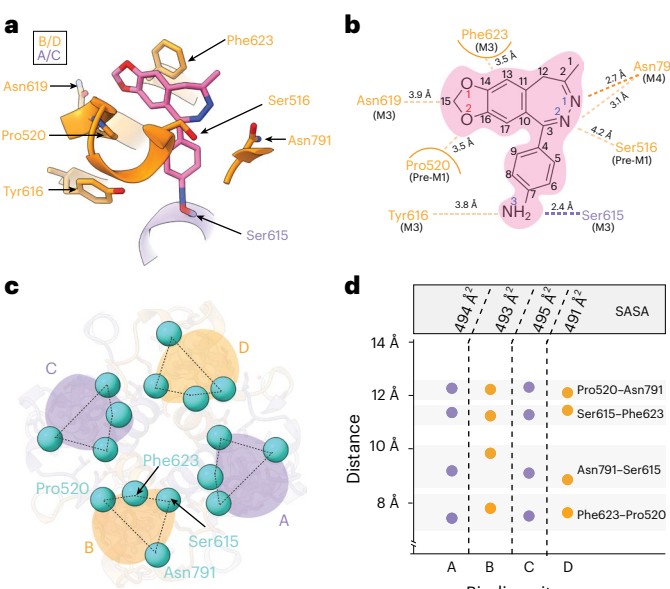

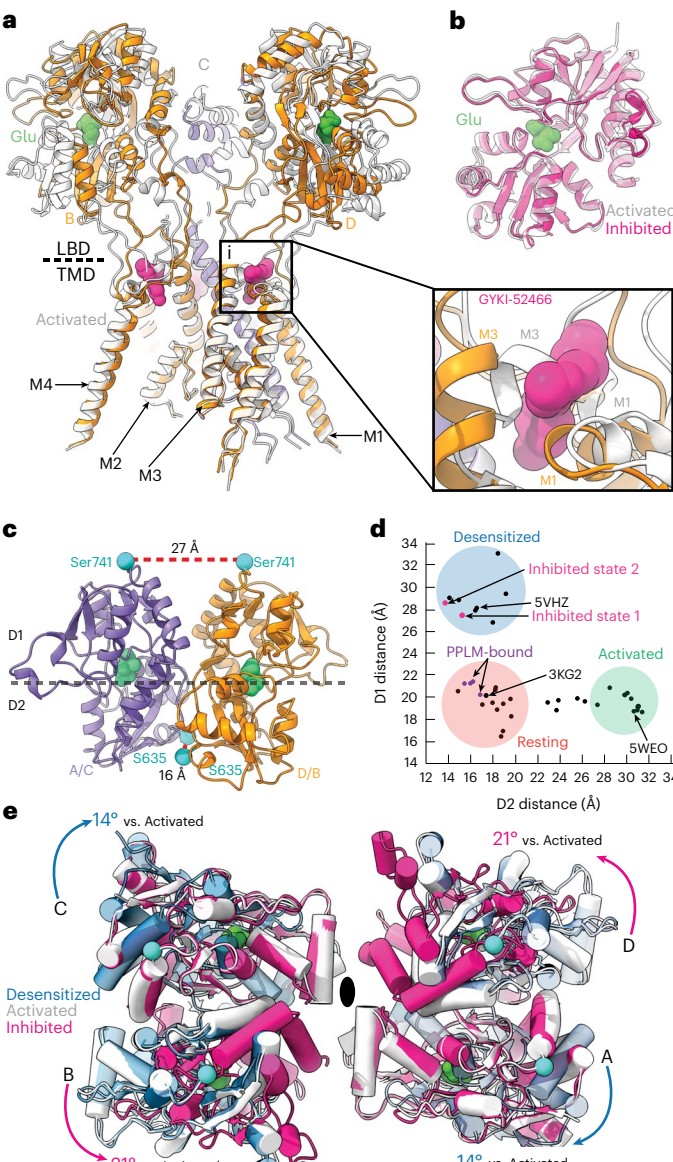

**Fig. 2 | High-resolution details of GYKI-52466 binding. a**, GYKI-52466 (pink, carbon; blue, nitrogen; red, oxygen) makes extensive contacts with residues within the TMD collar region (orange ribbons), including Ser615 on the neighboring M4 helix counterclockwise from the 'bound' subunit (purple). **b**, Schematic representation of the interactions between GYKI-52466 and TMD collar residues. The TM helix that each residue comes from is labeled below each residue. The π bonds are represented by curved lines; van der Waals interactions are represented by dashes. Carbon, nitrogen and oxygen atoms are colored according to **a**. **c**, Top-down view of the inhibited TMD, showing landmark residues based on the GYKI-52466-binding site. **d**, Plot detailing the inter-residue distances between landmark GYKI-52466-binding pocket residues and the solvent-accessible surface surrounding GYKI-52466.

interface with smFRET using a full-length GluA2(Gln)$_{flip}$ construct. To introduce specific labeling for the smFRET measurements, we substituted the free Cys residues to Ser and introduced the Leu467Cys

that they assume in the active state (Fig. 3e). Therefore, like their role in activation, the B and D subunits drive inhibition. We expect that, because the M3 helix kink is prevented by GYKI-52466 in the B and D subunits (Fig. 3a), this drives rearrangement in the LBD by the same subunits to accommodate inhibition. In contrast to the LBD layer, the GluA2-γ2$_{IS-1}$ TMD is markedly like the desensitized AMPAR TMD (r.m.s.d. = 0.7 Å; Extended Data Fig. 2c).

**Allosteric competition to control the AMPAR LBD**

The presence of GYKI-52466 in the channel collar region prevents the active-state transition during Glu binding, which prevents CTZ binding in the LBD[21]. Our structural data provide a direct mechanism of how PPLMs outcompete CTZ to allosterically control AMPAR function, which has been a long-standing mystery in the field[9,40,42–45]. Despite binding at disparate sites, we surmised that inhibition by GYKI-52466 likely has a greater effect on AMPARs because the inhibition mechanism directly ruptures the CTZ-binding pocket, while positive allosteric modulation by CTZ does not preclude GYKI-52466 binding[10]. We refer to this as allosteric competition.

Indeed, LBD dimers in AMPARs that are undergoing allosteric inhibition by GYKI-52466 (Fig. 4a; GluA2-γ2$_{IS-1}$) and positive allosteric modulation by CTZ (Fig. 4b; PDB 5WEO) are dramatically different. Two CTZ molecules act as a molecular glue between LBDs during positive allosteric modulation, maintaining a close distance between Ser741 pairs (Fig. 4b). However, during negative allosteric modulation, the 27 Å distance between Ser741 pairs ruptures the CTZ-binding site (Fig. 4a) and the D2–D2 separation is reduced to 15 Å from 31 Å (Fig. 4a,b).

To test the effects of allosteric modulation independently of the GluA2-γ2$_{EM}$ construct, we directly assayed the separation of the D1–D1

**Fig. 3 | Mechanism of allosteric inhibition. a**, Overlay of the allosterically inhibited state (orange/purple) and the activated state (white, PDB 5WEO; activated with 1 mM Glu + 100 μM CTZ). Inset: Close-up view of the GYKI-52466-binding site, revealing a steric clash with the kinked M3 helix found in the open state. **b**, Overlay of isolated LBD clamshells from the allosterically inhibited state (pink) and the open state (white). **c**, Local clamshell dimers within the LBD layer viewed from behind, showing the relative distances of the D1 and D2 lobes of the LBD dimer, as illustrated by the landmark residues Ser741 (D1) and Ser635 (D2). **d**, Plot of the D1 distance (Cα of Ser741) versus D2 distance (Cα of Ser635) measured for representative AMPAR structures captured in the resting, activated or desensitized state. On the basis of these measurements, the allosterically inhibited states (pink) cluster most closely with the desensitized state structures. **e**, Overlay of the overall LBD layer of the inhibited state (pink), activated state (white, PDB 5WEO) and desensitized state (blue, PDB 5VHZ) viewed from the top or extracellular side. Movements are measured within LBD dimers and mapped into the tetramer. The black oval marks the symmetry axis. Desensitization causes a 14° clockwise rotation of the A and C subunits within the LBD layer relative to the activated state[18]. In contrast, allosteric inhibition drives a counterclockwise rotation of the B and D subunits within the LBD layer relative to the active state.

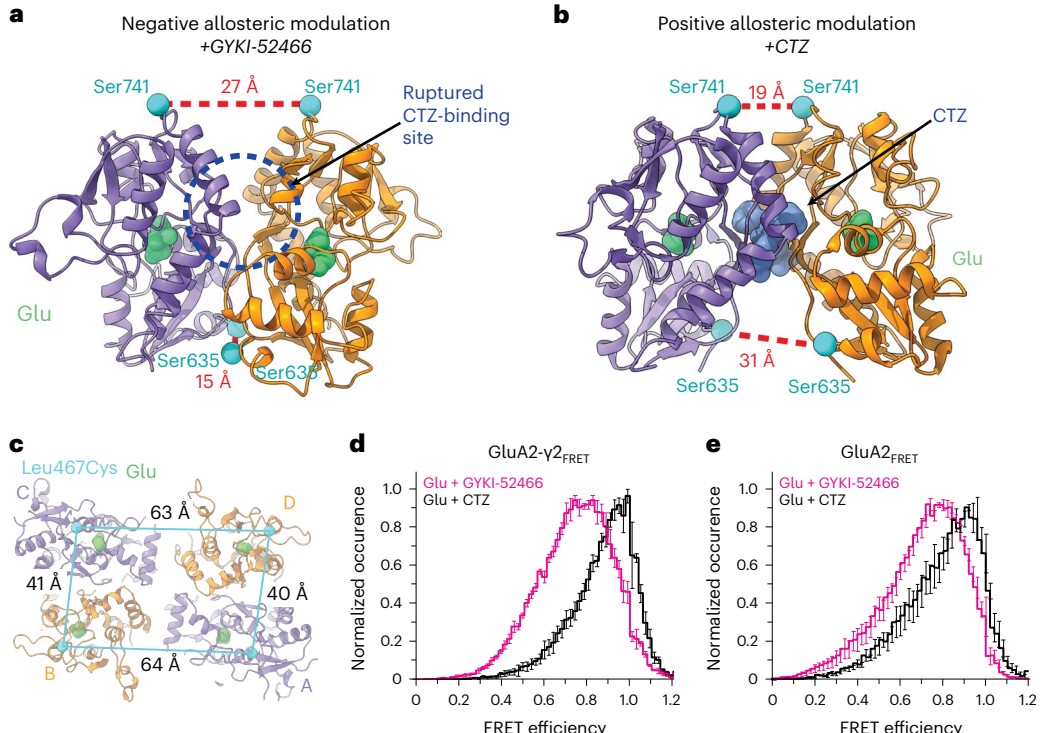

**Fig. 4 | Allosteric competition in AMPARs. a,** Local LBD clamshell from the allosterically inhibited state (GluA2-γ2_IS-1) with landmark residues to show D1 and D2 separation within the dimer. This represents the dimer during negative allosteric modulation. **b,** Local LBD clamshell dimer activated in the presence of CTZ (PDB 5WEO), showing decreased D1 separation and increased D2 separation. This is indicative of positive allosteric modulation. CTZ is shown in blue. Distances were measured as in Fig. 3. **c,** Top-down view of the LBD layer in GluA2-γ2_IS-1 with Leu467, where the Leu467Cys substitution is used for maleimide dye labeling, marked with a blue sphere, and intersubunit Leu467–Leu467 Cα distances are labeled. **d,** Plot of FRET efficiency between LBD clamshells within a local dimer when GluA2-γ2_FRET was treated with 1 mM Glu + 100 μM CTZ alone versus 1 mM Glu + 100 μM GYKI-52466. **e,** Same as in **d** but with GluA2_FRET. Data are represented as the mean values ± s.e.m. across multiple days. The number of molecules included in the analysis for each condition is as follows: GluA2-γ2_FRET (CTZ, *n* = 76; GYKI-52466, *n* = 77) and GluA2_FRET (CTZ, *n* = 62*; GYKI-52466, *n* = 96). *In this case, 30 molecules with 1 mM Glu and 100 μM CTZ were obtained from Carrillo et al.[60].

substitution at the top of the GluA2 LBD D1 (GluA2_FRET) (Methods). In the GluA2-γ2_FRET construct, full-length TARPγ2 was fused to the C terminus of GluA2 from GluA2_FRET using a GGS linker (Methods). The Leu467Cys substitution enables attachment of a dye by maleimide chemistry and establishes FRET pairs at the top of the GluA2 D1 in the LBD (Fig. 4c) within local LBD dimers. Other possible FRET pair distances in the tetramer occur at longer distances and do not contribute notable FRET[60]. The FRET efficiency when GluA2 is in the activated state (Glu + CTZ) is expected to be ~92% within an LBD dimer and ~19% across dimer pairs when Alexa-555 and Alexa-647 are used as the donor–acceptor pair.

We tested coupling of the D1 interface in GluA2-γ2_FRET during positive allosteric modulation in the presence of both 1 mM Glu and 100 μM CTZ (Fig. 4d), where the D1 lobes between LBD dimer pairs are at their closest[60-62] (Fig. 4b). The Glu and CTZ smFRET efficiency histogram showed higher efficiency than allosterically inhibited receptors (1 mM Glu and 100 μM GYKI-52466; Fig. 4d). This indicates that the distance across the D1 interface is shorter in the presence of the positive modulator CTZ than in the presence of the negative modulator GYKI-52466. To confirm that the decrease in smFRET efficiency in inhibitory conditions is not TARP dependent, we also tested smFRET efficiency in GluA2 homotetramers in the absence of TARPγ2 with the GluA2_FRET construct (Fig. 4e). Comparison of the GluA2_FRET and GluA2-γ2_FRET responses revealed similar effects of positive allosteric modulation (1 mM Glu + 100 μM CTZ) and allosteric inhibition (1 mM Glu + 100 μM GYKI-52466), which points to the decrease in smFRET efficiency not being TARP dependent but GYKI-52466 dependent or CTZ dependent.

The individual smFRET traces showed that the protein occupies 2–3 FRET efficiency states (Extended Data Fig. 8 and Methods). Using the highest-occurring state in GluA2-γ2_FRET, we obtained a FRET efficiency of 0.93 in the presence of CTZ and 0.82 in the presence of GYKI-52466 (Extended Data Fig. 8). These FRET efficiencies correspond to distances of 33 Å and 39 Å, respectively. The distance change of 6 Å agrees with our GluA2-γ2_IS-1 and GluA2-γ2_IS-2 cryo-EM structures, which show a D1–D1 (Leu467) distance change of 6 Å when compared to the CTZ-bound, activated-state AMPAR structure[18]. Thus, separation of the D1 lobes in AMPAR LBD dimers appears to be because of negative allosteric modulation by GYKI-52466. The lower FRET efficiency suggests additional conformations that are more decoupled at the D1–D1 interface than reconstructed with cryo-EM. These decoupled states are expected to be more dynamic and may not be homogeneous enough to classify into distinct cryo-EM classes (Extended Data Fig. 3a).

Collectively, our data suggest that negative allosteric modulation and positive allosteric modulation occupy different conformational states in the presence of Glu. The differences between the conformational spaces are a potential mechanism for allosteric competition between the two modulators (Fig. 4a,b). These data agree with our electrophysiological findings that the allosteric inhibition of GYKI-52466 outcompetes the positive allosteric modulation by CTZ of GluA2-γ2_EM (Fig. 1b and Extended Data Fig. 1a,b).

**Free energy landscape of the LBD dimer interface**
We hypothesized that the desensitization and allosterically inhibited states occupy different conformations in the LBD layer because of distinct free energy minima accompanying each state. To test this, we

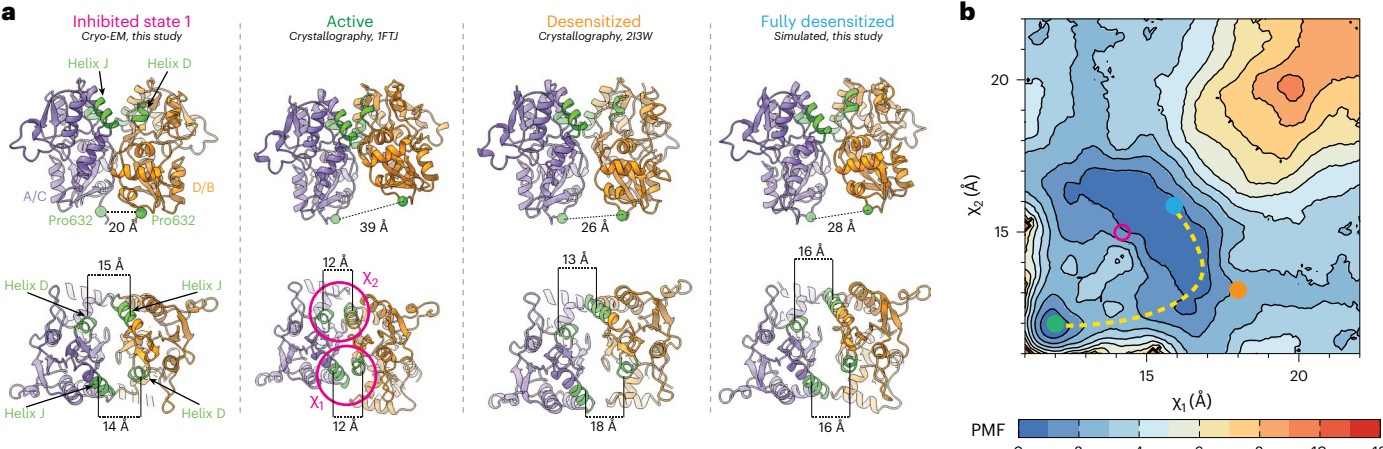

**Fig. 5 | Free energy landscapes governing desensitization and allosteric inhibition. a,** Top: rear views of local LBD clamshell dimers from the inhibited state, activated state, desensitized state or a projected maximally desensitized state. Bottom: top-down views of the allosterically inhibited, activated, desensitized or maximally desensitized states with helices D and J, colored green, labeled in accordance with their contribution to the two-dimensional order parameter. **b,** Free energy landscape governing LBD dimer conformations, that is, the separation of helices D and J at the dimer interface, with the measured distances of the activated (green), desensitized (orange) and maximally desensitized distances (blue) plotted on the diagram. Corresponding measurements from the allosterically inhibited state are shown in pink, which corresponds to where we would expect the allosterically inhibited state to sit within the free energy landscape. The dashed line suggests the most probable transition pathway between the active and desensitized conformations. The free energy landscape is contoured in increments of 1 kcal per mol.

computed a two-dimensional free energy landscape or potential of mean force (PMF) governing the rupture of a Glu-bound GluA2 LBD dimer interface using umbrella sampling free energy MD simulations (Methods).

Our PMF is a function of a two-dimensional order parameter ($\chi_1$ and $\chi_2$) that reports global changes within an LBD dimer. $\chi_1$ and $\chi_2$ describe the distances between the center of mass (COM) of helix J in D1 and the COM of helix D in D1 on a partner LBD in the dimer (Fig. 5a). ($\chi_1$, $\chi_2$) differs from the one-dimensional collective variable previously used to examine LBD dimer stabilities in AMPARs and kainate receptors through steered MD simulations[63]. While the LBDs are generally symmetric, the order parameter is not (Fig. 5a); $\chi_1$ describes the distance between helix pair J and D that is exterior facing, while $\chi_2$ describes the helix pair that faces the interior of the AMPAR in the context of a tetramer. Thus, this enables a two-dimensional approach to characterizing global changes in the LBD dimers.

Through sampling $\chi_1$ and $\chi_2$ in the context of Glu-bound LBDs, we can understand the energetics associated with rupturing the D1–D1 interface. Conformers for the umbrella sampling windows were generated using targeted MD simulations initiated with an activated GluA2 LBD and using a desensitized GluA2 LBD as a guide (Fig. 5a and Methods)[64,65]. Sampling windows were 1 Å increments along $\chi_1$ and $\chi_2$. The activated-state LBD dimer occupies a small free energy basin within the PMF, whereas the fully desensitized LBD occupies a substantially larger basin (Fig. 5b). The crystallized desensitized LBD, stabilized by a disulfide bond, lies near the most probable transition pathway between the active and desensitized conformations. This pathway suggests that, during rupture of the dimer interface, one J–D helix pair breaks before the other rather than both pairs breaking simultaneously, thereby circumventing a free energy barrier separating the two basins. The broader free energy basin associated with desensitization compared to activation may account for how short lived the active state is compared to the longer-lived desensitized state.

A point substitution, Leu483Tyr in helix D, was identified to strongly stabilize the nondesensitized (active) state[29]. To test whether our umbrella sampling strategy could recapitulate the effect of this substitution, we performed an analogous free energy calculation using the GluA2–Leu483Tyr LBD dimer. Umbrella sampling window conformers were generated from the crystal structure of the Leu483Tyr

LBD dimer[21]. The PMF of this nondesensitizing mutant revealed a substantially reduced free energy basin for the desensitized state, transforming the active-state basin into the global free energy minimum (Extended Data Fig. 9).

In inhibition, we observed separation of $\chi_1$ and $\chi_2$ compared to the activated LBD dimer (Fig. 5a). Interestingly, this state likely occupies a PMF basin that is distinct from the pathway of desensitization (Fig. 5b). This supports the observation that inhibition is similar but distinct from desensitization. The two-dimensional order parameter that we sampled in this experiment accounts for how the LBDs within a dimer pivot away from each other to accommodate D1 separation. We hypothesize that the distinct free energy basins of inhibited and desensitized LBDs account for the differences between allosteric inhibition and desensitization (Fig. 3e).

## Discussion

PPLMs bind to the AMPAR TMD and inhibit AMPARs by shunting the receptor into a distinct allosterically inhibited state following Glu binding (Fig. 6a), thereby decoupling Glu binding from channel opening. The inhibited states (GluA2-$\gamma2_{IS-1}$ and GluA2-$\gamma2_{IS-2}$) show marked differences compared to AMPAR structures bound to PPLMs in the resting state (Extended Data Fig. 2b). Previous studies suggested a two-step mechanism of inhibition, where an initial binding event by PPLMs is insufficient to produce complete inhibition[66,67]. Our results indicate a two-step mechanism involving GYKI-52466 binding followed by Glu binding in the LBDs and rupturing of the D1 interface between LBD dimers. This demonstrates how binding of PPLMs in the ion channel collar allosterically controls the AMPAR LBDs (Fig. 6a).

Our proposed mechanism bridges the electrophysiological studies of the competition between PPLM and CTZ with binding-site identification[9–11,19,44,46–49]. The GYKI-52466-binding site is consistent with mutagenesis studies conducted in the PPLM-binding pocket[8] and points to the likely involvement of Asn619 in stabilizing GYKI-52466 specifically. While the residues that coordinate GYKI-52466 are largely conserved across AMPAR subunits (Extended Data Fig. 10), the high-resolution details outlined here and identification of the negative allosteric modulation mechanism will improve small-molecule design in future studies.

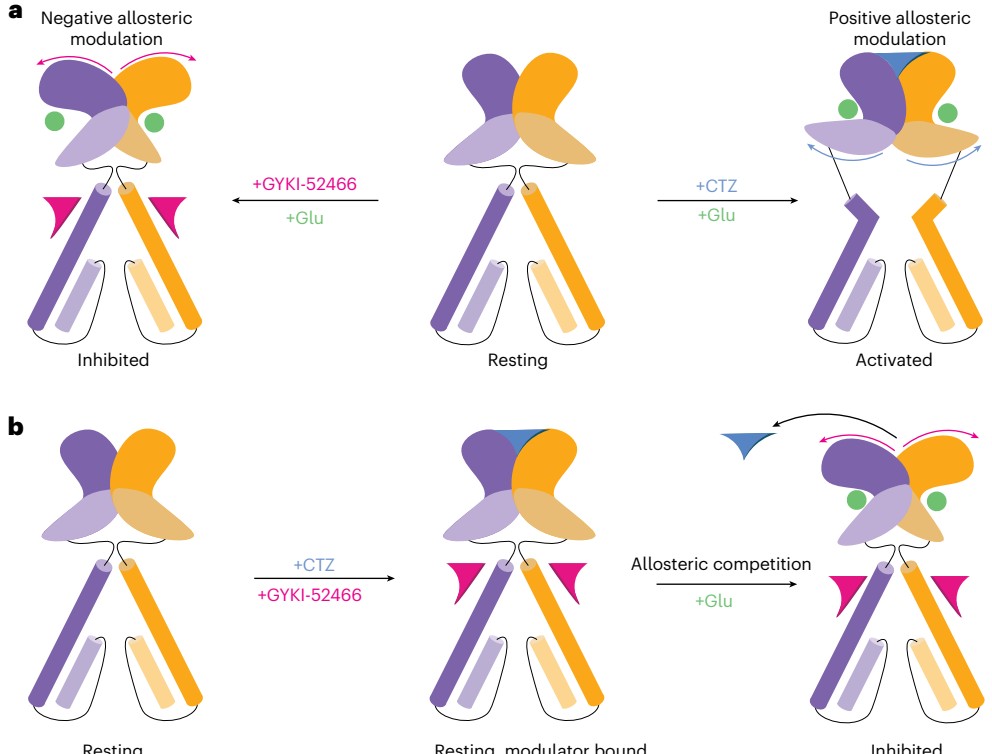

**Fig. 6 | Allosteric landscape of AMPARs. a**, Activation of AMPARs in the presence of the negative allosteric modulator GYKI-52466 produces the allosterically inhibited state, which we show in this study. In contrast, activation of AMPARs in the presence of the positive allosteric modulator CTZ produces the activated state. **b**, When both CTZ and GYKI-52466 are present, both can bind to the resting state of the receptor; however, after Glu binding, GYKI-52466 outcompetes CTZ to control the AMPAR LBD, resulting in CTZ being displaced and inhibition of the receptor.

Inhibition being a similar but distinct mechanism to desensitization also helps conceptualize future therapeutics targeting AMPARs. The motion of the domains in the B and D subunits that accompanies inhibition (Fig. 3e) may provide a route for specificity in small-molecule targeting considering that these positions are enriched for specific GluA subunits in native AMPARs[52–54].

The competition between positive (for example, CTZ) and negative (for example, PPLMs) allosteric modulators accounts for how GYKI-52466 and CTZ produce opposing effects on channel conductance[8–10,43]. Early studies postulated a shared binding site for GYKI-52466 and CTZ because of their countervailing effects on AMPAR channel conductance[43,68]. However, CTZ and PPLMs act at distinct sites[8,21,40], thereby rendering their mechanistic competition unclear. Our data agree with previous findings that PPLMs can outcompete positive allosteric modulators that bind to disparate sites such as CTZ. However, our data expand on this idea by providing insight into how this competition is achieved. Both PPLMs and CTZ can bind to resting-state AMPARs[8,18,41,49,66,69,70]. Our data reveal that the competition mechanism is, therefore, dependent on the presence of Glu and negative allosteric modulation by GYKI-52466 prevents CTZ from positively modulating AMPARs through rupturing the CTZ-binding site (Fig. 6b).

Inhibition by PPLMs appears to be independent of TARPs. However, noncompetitive inhibition of AMPARs may function similarly across different drug types. AMPARs are tightly regulated by TARPs and other auxiliary subunits[27,28,41,47,55,56] and recently identified compounds (for example, JNJ-55511118, JNJ-118, JNJ-059 and LY-481) demonstrate selectivity for particular AMPAR–TARP complexes[53,71–76]. The binding sites are distinct from those of PPLMs, located within the interface between TARPs and AMPARs. It is possible that these TARP-dependent noncompetitive inhibitors act similarly to PPLMs. Resolving this question will require additional studies with

AMPARs activated in the presence of TARP-dependent noncompetitive inhibitors.

In sum, we reveal how AMPARs are allosterically inhibited by PPLMs and how allosteric competition occurs within AMPARs. Our data provide a foundation for structure-based drug design against AMPARs, as well as a framework to study allostery across iGluRs.

## Online content

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

## Methods

### Construct design

The fusion construct GluA2-γ2$_{EM}$ was published previously and extensively validated for functional assays and structure determination[18,26,47–49]. The GluA2 subunit in the construct is adapted from the GluA2* construct, where the ATD–LBD linker is truncated and glycosylation sites are substituted. More specifically, rat GluA2$_{flip}$ (NP_058957) had 36 residues truncated at the C terminus after TM4, 6 residues truncated from the ATD–LBD linker (Leu378, Thr379, Leu381, Pro382, Ser383 and Gly384), and N-linked glycosylation sites substituted and knocked out (Asn235Glu, Asn385Asp and Asn392Gln). Gln was introduced at the Gln/Arg site (Arg586Gln) to stabilize the tetrameric form of the receptor[17]. More details on GluA2* can be found in Yelshanskaya et al.[22], where it was functionally validated and used for structure determination. GluA2* was directly fused to the N terminus of mouse TARPγ2 (NP_031609), which had its C terminus removed immediately after TM4 (truncated at Leu207). TARPγ2 residues Thr-Gly-Gly were introduced as spacers within a thrombin cleavage site (Leu-Val-Pro-Arg-Gly-Ser), which was followed by a C-terminal enhanced green fluorescent protein (eGFP) for monitoring expression, Strep Tag II and a stop codon. GluA2-γ2$_{EM}$ was inserted after the cytomegalovirus (CMV) promoter into the pEG BacMam vector for baculovirus-driven protein expression in mammalian cells[77]. GluA2-γ2$_{EM}$ was originally established and validated in Twomey et al.[47] and extensively used for later structural studies[18,26,47–49].

For smFRET, the GluA2$_{FRET}$ construct was designed in a pRK5 vector containing the CMV promoter as previously described[60]. Briefly, free Cys residues 89, 196 and 436 in full-length, wild-type rat GluA2(Gln)$_{flip}$ were substituted to Ser and Cys was introduced at position 467 (Leu467Cys) for maleimide dye attachment to measure the intradimer interface of the LBD.

To generate the GluA2-γ2$_{FRET}$ construct for smFRET, TARPγ8 from the GluA2–TARPγ8 fusion construct containing a Leu467Cys site in GluA2 (ref. 60; GluA2$_{FRET}$ above) was replaced with full-length mouse TARPγ2 from the GluA2–TARPγ2 construct[62] using restriction enzyme cloning with restriction enzymes BamHI and EcoRV to generate GluA2–TARPγ2 with Cys467 in GluA2.

### Protein expression and purification

The GluA2-γ2$_{EM}$ bacmid was prepared as previously described[18,26,47]. P1 baculovirus was generated by transfecting ExpiSf9 cells (Gibco, A35243) cultured at 27 °C with polyethyleneimine (molecular weight, 40,000; PolyScience, 24765). After 5 days, P1 virus was harvested and expression in mammalian cells was induced by the addition of P1 baculovirus to Expi293F GnTI⁻ cells (Gibco, A39240) grown in Expi293 medium (Gibco, A14135101) in a 1:10 ratio of P1 virus to culture volume. Cells were grown at 37 °C in 5% CO$_2$. Then, 12–24 h after induction, the cell culture medium was brought up to 10 mM sodium butyrate (Sigma, 303410) and 2 μM ZK 20075 (Tocris, 2345) and moved into a 30 °C, 5% CO$_2$ incubator. The cells were harvested 72 h after transduction by centrifugation (5,000g, 20 min at 4 °C), washed with PBS (pH 7.4) with protease inhibitors added (0.8 μM aprotinin, 2 μg ml⁻¹ leupeptin, 2 μM pepstatin A and 1 mM phenylmethylsulfonyl fluoride) and then pelleted again (4,800g, 10 min at 4 °C). The supernatant was discarded and pellets were stored at −80 °C until purification. Pellets were thawed rotating in lysis buffer (150 mM NaCl and 20 mM Tris pH 8.0) with protease inhibitors added. Cells were lysed in an ice bath with a blunt probe sonicator (three cycles, 1 s on, 1 s for 1 min, 20 W power). Lysed cells were centrifuged to pellet large cellular debris (4,800g, 20 min at 4 °C). The supernatant was ultracentrifuged to pellet membranes (125,000g, 45 min), which were solubilized in solubilization buffer (150 mM NaCl, 20 mM Tris pH 8.0, 1% n-dodecyl-β-D-maltopyranoside (DDM; Anatrace, D310)) and 0.2% cholesteryl hemisuccinate Tris salt (Anatrace, CH210) for 2 h at 4 °C under constant stirring. Insoluble material was pelleted in an ultracentrifuge (125,000g, 45 min at 4 °C) and solubilized protein

was incubated with 0.75 ml of Strep-Tactin XT 4Flow resin (IBA, 2-5010) per 1 l of cells overnight, rotating at 4 °C. The following day, the resin was collected by gravity flow and washed with 20 column volumes of glyco-diosgenin (GDN) buffer (150 mM NaCl, 20 mM Tris pH 8.0 and 0.01% GDN (Anatrace, GDN101)), before elution in GDN buffer made up to 50 mM D-biotin. Eluate was collected in a centrifugal concentrator and concentrated into a 500-μl volume at 4 °C. To remove eGFP and Strep Tag II, the concentrated protein was incubated with thrombin (1:200 w/w) for 1 h at 22 °C. The cleavage reaction was separated over a Superose 6 increase 10/300 column (Cytiva, 29091596) using an AKTA fast protein liquid chromatograph in GDN buffer. Peak fractions were collected and concentrated to 4.5 mg ml⁻¹.

### Sample preparation and data collection

UltrAuFoil 300 mesh R 1.2/1.3 grids (Electron Microscopy Services, Q350AR13A) were plasma-treated in a Pelco Easiglow (25 mA, 120 s glow time and 10 s hold time; Ted Pella, 91000). Purified sample was split into two conditions. The IS-1 sample was made up to 100 μM CTZ (Tocris, 07-131-0) and spun in an ultracentrifuge to pellet insoluble material before the preparation of grids (75,000g, 45 min), whereas the IS-2 sample was made up to 100 μM CTZ and 100 μM GYKI-52466 (Tocris, 1454) before centrifugation (75,000g, 45 min). IS-1 samples were spiked with 100 μM GYKI-52466 and 1 mM Glu (pH 7.4) immediately before application to grids. IS-2 samples were only spiked with 1 mM Glu before application to grids. In both cases, 3 μl of sample was applied to glow-discharged grids in an FEI Vitrobot Mark IV (Thermo Fisher Scientific; wait time, 10 s; blot force, 5; blot time, 4 s) at 8 °C and 100% humidity and plunge-frozen in liquid ethane. Grids were imaged with a 300-kV Titan Krios 3i microscope equipped with fringe-free imaging, a Falcon 4i camera and a Selectris energy filter set to a 10-eV slit width. Micrographs were collected with a dose rate of 8.15 e⁻ per pixel per s and a total dose of 40.00 e⁻ per Å². We collected 8,800 micrographs of the GYKI-1 condition (0.93 Å per pixel) and 7,900 micrographs of the GYKI-2 condition (0.93 Å per pixel). Automated collection was achieved with EPU software from Thermo Fisher Scientific.

### Image processing

Cryosparc[78] was used for all aspects of image processing (refer to Extended Data Figs. 3 and 4 for details). The reconstruction quality was tested for anisotropic contribution to the Fourier shell correlation (FSC) with 3DFSC[79].

### Model building, refinement and structural analysis

Molecular modeling, refinement and analysis were performed with a combination of ChimeraX[80], ISOLDE[81], Coot[82] and PHENIX[83,84] made accessible through the SBgrid consortium[85]. As a starting model, the activated state of GluA2-γ2$_{EM}$ (refs. 18,48) (PDB 5WEO) was used. Each domain (ATD, LBD and TMD) was isolated and underwent rigid-body fitting into the GluA2-γ2$_{IS-1}$ full-length cryo-EM reconstruction using ChimeraX. The rigid-body position of each protomer was refined by isolating it within the domain and rigid-body fitting. Then, each domain was joined into a single model. The exact positioning of each amino acid was fine-tuned on the basis of the locally refined map of each domain using Coot. Then, ISOLDE was used to refine the model and GYKI-52466 was placed in the map with Coot and merged into the model. PHENIX was used to refine the final model. To model GluA2-γ2$_{IS-2}$, the GluA2-γ2$_{IS-1}$ model underwent rigid-body fitting into the GluA2-γ2$_{IS-2}$ reconstruction and refined with ISOLDE and PHENIX. Model quality was assessed with MolProbity[86]. Visualizations and domain measurements were performed in ChimeraX. Pore measurements were made with MOLE Online[87].

### Labeling, acquisition and analysis for smFRET

HEK293T cells (American Type Culture Collection (ATCC), CRL-3216) overexpressing GluA2$_{FRET}$ or GluA2-γ2$_{FRET}$ receptors were labeled with

1:4 ratio of maleimide derivatives of Alexa-555 (donor) and Alexa-647 (acceptor) fluorophores (Invitrogen) in extracellular buffer (135 mM NaCl, 3 mM KCl, 2 mM CaCl$_2$, 20 mM glucose and 20 mM HEPES pH 7.4) at room temperature for 30 min. After labeling, the cells were washed and solubilized for 1 h at 4 °C with buffer containing 1% lauryl maltose neopentyl glycol (Anatrace), 2 mM cholesteryl hydrogen succinate (CHS; MP Biomedicals) and ¼ protease inhibitor tablet (Pierce) in PBS. Solubilized cells were filtered from insoluble debris by ultracentrifugation at 100,000$g$ for 1 h at 4 °C using a TLA 100.3 rotor.

For the slide preparation, we followed established experimental methods as previously described[88–92]. The coverslips were initially cleaned by bath sonication in Liquinox phosphate-free detergent (Fisher Scientific) and acetone treatment. Further cleaning involved incubating the slides in a 4.3% NH$_4$OH and 4.3% H$_2$O$_2$ solution at 70 °C, followed by plasma cleaning using a Harrick Plasma PDC-32G Plasma Cleaner. The cleaned glass was aminosilanated using Vectabond reagent (Vector Laboratories), followed by polyethylene glycol (PEG) treatment with 0.25% w/w 5 kDa biotin-terminated PEG (NOF Corporation) and 25% w/w 5 kDa mPEG succinimidyl carbonate (Laysan Bio), followed by a secondary PEG treatment with 25 mM short-chain 333 Da MS(PEG)4 methyl-PEG-NHS-ester reagent (Thermo Scientific). A microfluidics chamber was constructed on the slide, comprising an input port, a sample chamber and an output port. To coat the biotinylated surface with streptavidin molecules, 0.2 mg ml$^{-1}$ streptavidin in 1× smFRET imaging buffer (1 mM DDM, 0.2 mM CHS and 1× PBS) was introduced into the chamber and incubated for 10 min before washing with 1× PBS. Next, 60 µl of biotinylated goat anti-mouse IgG (H + L) secondary antibody at 2.7 ng µl$^{-1}$ (Jackson Immunoresearch Laboratories, cat. no. 115-065-003) in 1× PBS was flowed through the chamber and incubated for 20 min, before washing with 1× PBS.

Following this, either 60 µl of anti-GluR2 at 3 ng µl$^{-1}$ for GluA2$_{FRET}$ purification (clone L21/32; BioLegend) or 60 µl of anti-TARPγ2 at 2.4 ng µl$^{-1}$ for GluA2-γ2$_{FRET}$ purification (clone N245/36; Millipore) in 1× PBS was applied twice through the chamber and incubated for 20 min, followed by washing with 1× PBS. BSA (0.1 mg ml$^{-1}$) was introduced into the chamber and incubated for 15 min, before washing with 1× PBS. Detergent-solubilized purified proteins were attached to the glass slide using an in situ immunoprecipitation method by applying 50 µl of sample three times through the chamber and incubating for 20 min. Then, 90 µl of oxygen-scavenging solution buffer system (ROXS) was applied inside the chamber containing 1 mM methyl viologen, 1 mM ascorbic acid, 0.01% w/w pyranose oxidase, 0.001% w/v catalase, 3.3% w/w glucose (all from Sigma-Aldrich), 1 mM DDM (Chem-Impex) and 0.2 mM CHS (MP Biomedicals) in PBS pH 7.4. For the CTZ condition, 1 mM Glu and 100 µM CTZ were introduced into the ROXS. In the GYKI-52466-treated condition, 1 mM Glu and 100 µM GYKI-52466 (Millipore-Sigma) were introduced into the ROXS.

The smFRET data were collected using a MicroTime 200 Fluorescence Lifetime Microscope from PicoQuant. A donor excitation laser (532 nm; LDH-D-TA-530; Picoquant) and an acceptor excitation laser (637 nm; LDH-D-C-640; Picoquant) were used with a pulsed interleaved excitation scheme to excite the fluorophores. Emitted photons were collected through the objective lens (×100, 1.4 numerical aperture; Olympus). Emission filters for the donor (550 nm; FF01-582/64; AHF or Semrock) and acceptor (650 nm; 2XH690/70; AHF) were used to select photons for each detection channel. These photons were directed to two single-photon avalanche diodes (SPCM CD3516H; Excelitas Technologies) to measure the fluorescence intensity for each fluorophore. The donor and acceptor fluorescence intensities were recorded for one protein at a time.

In our data analysis, we selected only those molecules that exhibited a single photobleaching step in both the donor and the acceptor channels. This stringent criterion ensured that only one donor and one acceptor fluorophore were attached to each GluA2 protein. Furthermore, we retained only those molecules that displayed anticorrelation

between the donor and acceptor fluorescence, confirming that the fluorophores were engaged in FRET before photobleaching. Molecules not exhibiting these characteristics were excluded from the final analysis. The number of molecules included in the analysis for each condition was as follows: GluA2-γ2$_{FRET}$ (CTZ, $n$ = 76; GYKI-52466, $n$ = 77) and GluA2$_{FRET}$ (CTZ, $n$ = 62*; GYKI-52466, $n$ = 96). *In this case, 30 molecules with 1 mM Glu and 100 µM CTZ were obtained from Carrillo et al.[60].

The corrected donor and acceptor intensities over time were then used to calculate a FRET efficiency trace for each molecule. These traces were pooled for each condition and used to create FRET efficiency distribution histograms for each condition. We conducted step transition and state identification (STaSI) analysis to determine the number of conformational states in each condition[93]. The smallest number of states that accurately described the data as determined by the STaSI analysis was adopted as the final number of states for each condition. Using the results of the STaSI analysis and Origin software (OriginLab), the FRET efficiency histograms for each condition were fitted with Gaussian curves to represent the conformational states within the overall distributions.

To test for the statistical difference between conditions CTZ and GYKI-52466, the FRET efficiency mode was obtained for each day, as this more accurately represents the histogram peak. The mean and s.d. were calculated across these days. A two-sample $t$-test, assuming a one-tail distribution with known variances, was used to assess the statistical differences between the conditions using Origin software (OriginLab).

### Electrophysiology

For electrophysiological measurements of GluA2-γ2$_{EM}$, which contained eGFP for cell detection, 1 µg of DNA was transfected into HEK293T cells (ATCC, CRL-3216) in 3-cm culture dishes using Lipofectamine 2000. Patch-clamp recordings were performed 24–48 h after transfection using fire-polished borosilicate glass (Sutter Instrument). Pipettes with 1–4 MΩ resistance were filled with internal solution: 110 mM CsF, 30 mM CsCl, 4 mM NaCl, 0.5 mM CaCl$_2$, 10 mM HEPES and 5 mM EGTA (adjusted to pH 7.4 with CsOH). The extracellular solution consisted of 150 mM NaCl, 3 mM KCl, 2 mM CaCl$_2$ and 10 mM HEPES adjusted to pH 7.4 with NaOH. External solutions were locally applied to lifted cells or patches using an SF-77B perfusion fast-step (Warner Instruments). For inhibition concentration–response determination, 100 µM CTZ was preincubated in extracellular buffer for at least 30–60 s, along with the corresponding GYKI-52466 concentration. For channel activation, 1 mM Glu with 100 µM CTZ and the corresponding GYKI-52466 concentration was applied for 500 ms and recordings were allowed to reach equilibrium before obtaining 2–10 sweeps per condition for averaging. The mean of the residual current was obtained using a range between 200 and 500 ms after Glu application and used for inhibition concentration–response analysis. Recordings were performed using an Axopatch 200B amplifier (Molecular Devices) at −60 mV hold potential, acquired at 2 kHz using pCLAMP10 software (Axon 200B and Digidata 1550A; Molecular Devices). Individual patch-clamp traces and the average residual current for IC$_{50}$ were analyzed using Clampfit 11 software (Molecular Devices). The inhibition concentration–response results were analyzed using the Levenberg–Marquardt iteration algorithm for a nonlinear curve fit using OriginPro 2023b. The experimental data were fit with the following equation:

$$y = A_1 + \frac{A_2 - A_1}{1 + 10^{\left(\left(\log_{x_0 - x}\right)p\right)}}$$

The dataset was analyzed using the concatenate fit mode, ensuring a robust assessment of the concentration–response behavior. Representative traces were graphed, normalized and calculated using Origin software (OriginLab).

## Free energy MD simulations

The conformational free energy landscape or PMF of the LBD dimer was computed using umbrella sampling simulations. A two-dimensional order parameter ($\chi_1, \chi_2$) described the large-scale conformational transitions between each LBD of the dimer. $\chi_1$ and $\chi_2$ each indicated the distance between the COM of atoms N, CA, CB, C and O in residues 482–488, helix D, and the COM of the same atoms in residues 748–757, helix J. Helices D and J formed the dimer interface. Coordinates for the umbrella sampling windows were generated by targeted (biased-potential) MD simulations using CHARMMA[94] in 1 Å increments along $\chi_1$ and $\chi_2$. These coordinates were initiated from the crystal structure of a Glu-bound GluA2 LBD dimer (PDB 1FTJ)[64]. For GluA2–L483Y, these coordinates were initiated from the crystal structure of the mutant LBD dimer (PDB 1LB8)[21]. Missing residues were built using the ModLoop server[95] and missing residue side chains were built using SCWRL4 (ref. 96).

All simulations were performed using CHARMM36 with explicit solvent at 300 K. The all-atom potential-energy function PARAM27 for proteins[97,98] and the TIP3P potential-energy function for water[99] were used. Each simulation system contained ~56,000 atoms and 39 $Na^+$ and 47 $Cl^-$ ions were added to the bulk solution to give ~150 mM NaCl and an electrically neutral system. Periodic boundary conditions were used with an orthorhombic cell with approximate dimensions of 96 Å × 78 Å × 78 Å. Equilibration was carried out in the NVT ensemble with restraints applied to the backbone and sidechain atoms, which were slowly released over the course of the equilibration. Production simulations were carried out in the NPT ensemble at 1 atm and 300 K (ref. 100). Long-range electrostatic interactions were computed using the particle mesh Ewald algorithm[101].

The PMF comprised 140 umbrella sampling windows totaling 364 ns of simulation time and 398 ns for GluA2–L483Y. Harmonic biasing potentials with a force constant of 2 kcal per mol per Å centered on ($\chi_1, \chi_2$) were used. Each PMF was computed using the weighted histogram analysis method[102,103] to unbias and recombine the sampled distribution functions from all windows.

## Multiple sequence alignment

Rat Gria1–Gria4 protein sequences were accessed from UniProt (P19490, Gria1; P19491, Gria2; P19492, Gria3; P19493, Gria4) and aligned using the server-based Expresso implementation of T-Coffee[104,105]. The alignment was visualized using Jalview[106].

## Reporting summary

Further information on research design is available in the Nature Portfolio Reporting Summary linked to this article.

## Data availability

The accession codes for GluA2-$\gamma 2_{IS-1}$ and GluA2-$\gamma 2_{IS-2}$ are EMD-43275 and EMD-43276, respectively. The full maps (before local refinement and signal subtraction) are the primary cryo-EM maps in each deposition and each local map is supplied as a supplemental file in each deposition. The GluA2-$\gamma 2_{IS-1}$ and GluA2-$\gamma 2_{IS-2}$ structures are deposited to the PDB (8VJ6 and 8VJ7, respectively). Source data are provided with this paper.

## Code availability

All conformers from the MD simulation trajectories, data from umbrella sampling and analysis code are publicly available from Zenodo (https://doi.org/10.5281/zenodo.10967297)[107].

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

## Acknowledgements

We thank members of the Twomey, Huganir, Lau and Jayaraman labs for insightful discussions and L. Dillard (Twomey lab) for assistance with collecting the GluA2-$\gamma2_{IS-2}$ dataset. We thank M. Catipovic (JHU) for insightful comments on the paper. All cryo-EM data were collected at the Beckman Center for Cryo-EM at Johns Hopkins with assistance from D. Sousa and D. Ding. Computational resources were provided by the Maryland Advanced Research Computing Center (MARCC) and Advanced Research Computing at Hopkins (ARCH) at Johns Hopkins University. E.C.T. is supported by the Searle Scholars Program (Kinship Foundation 22098168) and the Diana Helis Henry Medical Research Foundation (142548). R.L.H. is supported by National Institutes of Health (NIH) grant R37 NS036715. A.Y.L. is supported by NIH grant R01 GM094495. V.J. is supported by NIH grant R35 GM122528. C.U.G. is supported by NIH grant F99 NS130928. W.D.H. is supported by NIH grant K99 MH132811.

## Author contributions

E.C.T. and R.L.H. supervised all aspects and planning of this research. E.C.T. and W.D.H. designed the project. E.C.T. and W.D.H. wrote the paper with input from all authors. W.D.H. prepared samples for cryo-EM, collected, processed and analyzed the cryo-EM data and built models with E.C.T. A.M.R. assisted with protein expression, model building, data analysis, structural analysis and uncovering the inhibition mechanism with W.D.H. V.J. and C.U.G. designed the smFRET and electrophysiology experiments with input from E.C.T. and W.D.H. C.U.G. carried out the smFRET and electrophysiology experiments under the supervision of V.J. C.U.G. carried out the statistical analysis of smFRET and electrophysiology data under the supervision of V.J. A.Y.L. planned and carried out all MD simulation studies and analysis.

## Competing interests

R.L.H. is a scientific cofounder and scientific advisory board (SAB) member of Neumora Therapeutics and an SAB member of MAZE Therapeutics. The other authors declare no competing interests.

## Additional information

**Extended data** is available for this paper at https://doi.org/10.1038/s41594-024-01328-0.

**Correspondence and requests for materials** should be addressed to Albert Y. Lau, Richard L. Huganir or Edward C. Twomey.

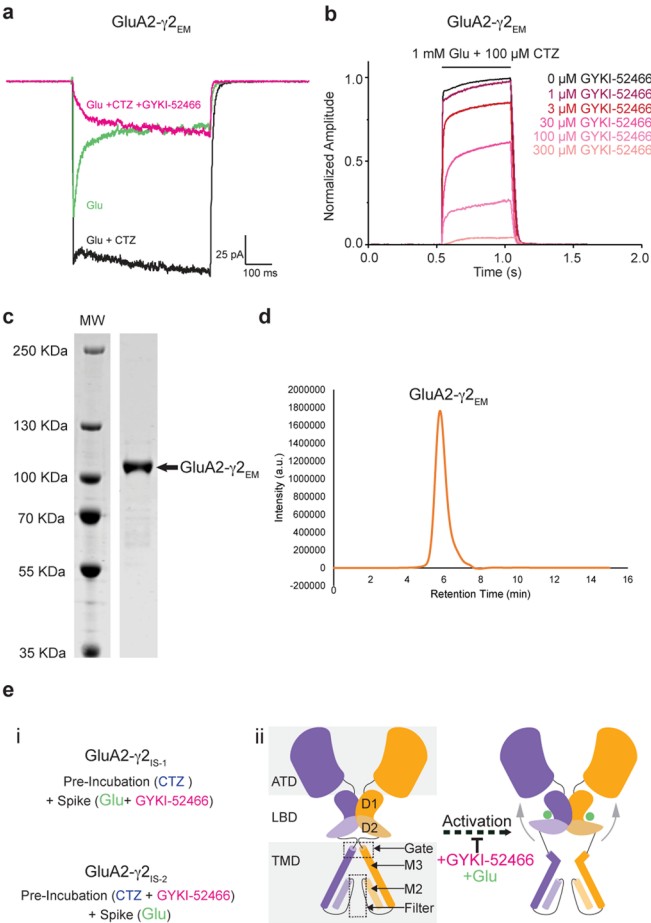

**Extended Data Fig. 1 | Electrophysiology, purification, and preparation of GluA2-γ2$_{EM}$.** (**a**) Representative whole-cell patch clamp traces from HEK293T cells expressing GluA2-γ2$_{EM}$ in the presence of either 1 mM Glu, 1 mM Glu + 100 μM CTZ, or 1 mM Glu + 100 μM CTZ + 100 μM GYKI-52466. Traces representative of at least three individual cells. (**b**) Representative normalized whole-cell patch clamp traces of HEK293T cells expressing GluA2-γ2$_{EM}$ treated with 1 mM Glu + 100 μM CTZ either alone or with increasing concentrations of GYKI-52466. For each concentration, data were obtained from at least three different cells. (**c**) Coomassie-stained SDS-PAGE gel of purified GluA2-γ2$_{EM}$ sample showing a single band at the predicted molecular weight (arrow). (**d**) Size exclusion chromatogram of purified GluA2-γ2$_{EM}$ sample showing a single monodispersed peak at the predicted retention time for a GluA2-γ2$_{EM}$. (**e**) (i) Treatment regimens for producing the different inhibited states GluA2-γ2$_{IS-1}$ and GluA2-γ2$_{IS-2}$, (ii) cartoon demonstrating the targeted outcome of activating GluA2-γ2$_{EM}$ in the presence of the inhibitor GYKI-52466.

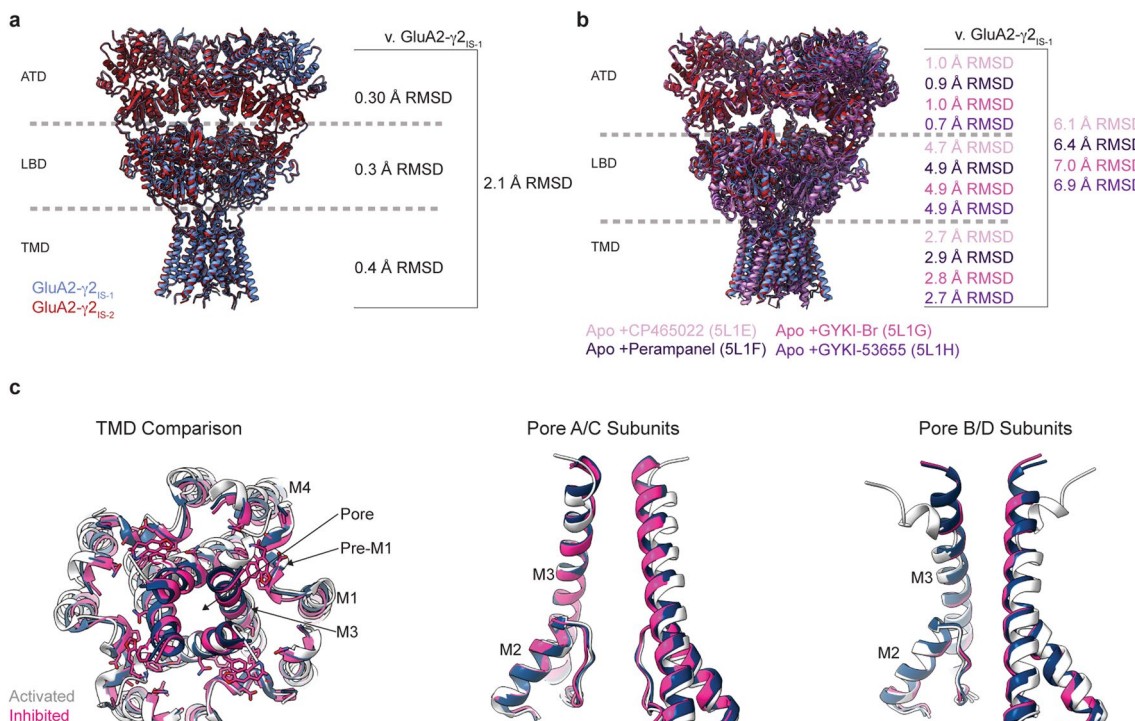

**a**

ATD

0.30 Å RMSD

LBD

0.3 Å RMSD

2.1 Å RMSD

TMD

0.4 Å RMSD

v. GluA2-γ2$_{IS-1}$

GluA2-γ2$_{IS-1}$
GluA2-γ2$_{IS-2}$

**b**

ATD

1.0 Å RMSD
0.9 Å RMSD
1.0 Å RMSD
0.7 Å RMSD

LBD

4.7 Å RMSD
6.4 Å RMSD
4.9 Å RMSD
4.9 Å RMSD

TMD

2.7 Å RMSD
2.9 Å RMSD
2.8 Å RMSD
2.7 Å RMSD

v. GluA2-γ2$_{IS-1}$

6.1 Å RMSD
6.4 Å RMSD
7.0 Å RMSD
6.9 Å RMSD

Apo +CP465022 (5L1E)   Apo +GYKI-Br (5L1G)
Apo +Perampanel (5L1F)   Apo +GYKI-53655 (5L1H)

**c**

TMD Comparison

M4
Pore
Pre-M1
M1
M3

Activated
Inhibited
Desensitized

Pore A/C Subunits

M3
M2

Pore B/D Subunits

M3
M2

**Extended Data Fig. 2 | Comparison between inhibited states and AMPAR structures in resting state bound to PPLMs.** (**a**) overlay of GluA2-γ2$_{IS-1}$ and GluA2-γ2$_{IS-2}$ from this study demonstrating very minor deviation in the two states from each other. (**b**) overlay of GluA2-γ2$_{IS-1}$ against crystal structures of GluA2 in complex with PPLMs: CP465022 (PDB: 5L1E), Perampanel (PDB: 5L1F), GYKI-Br (PDB: 5L1G), and GYKI-53655 (PDB: 5L1H). RMSD is greatest within the LBD layer,

in agreement with the conformational shifts observed following GluA2-γ2$_{EM}$ activation in the presence of GYKI-52466. (**c**) Overall comparison of TMDs, A/C subunit ion channel helices, and B/D subunit ion channel helices between activated (PDB 5WEO, white), inhibited (GluA2-γ2$_{IS-1}$, pink) and desensitized (PDB 7RYY, blue). Compared to the entire GluA2-γ2$_{IS-1}$ TMD, the activated state TMD has 1.0 Å RMSD, and desensitized state TMD 0.7 Å RMSD.

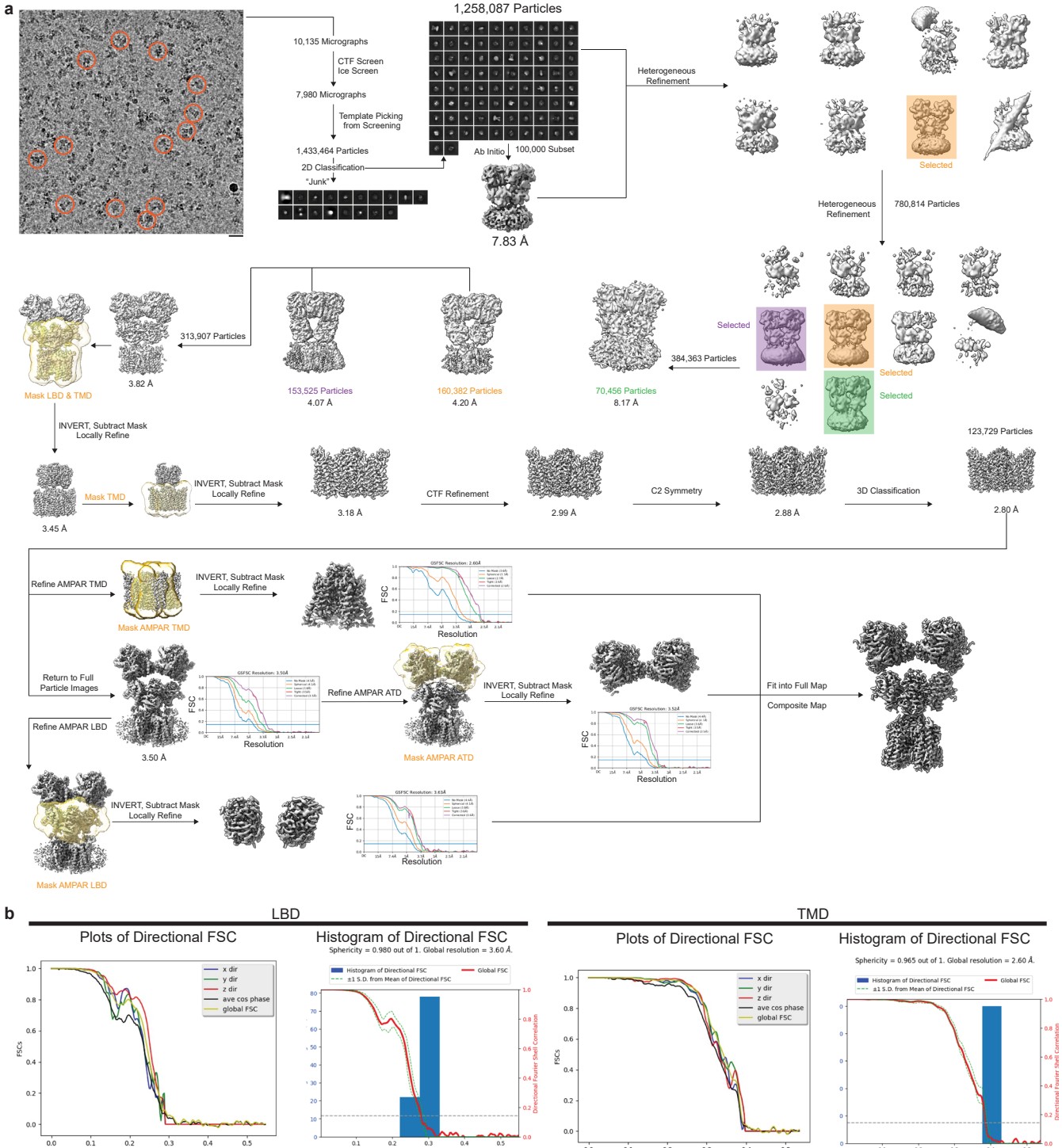

**Extended Data Fig. 3 | Cryo-EM processing workflow for GluA2-γ2$_{IS-1}$.** (**a**) Image processing workflow and approach in Cryosparc. (**b**) Three-dimensional Fourier shell correlation analysis for local LBD and TMD maps.

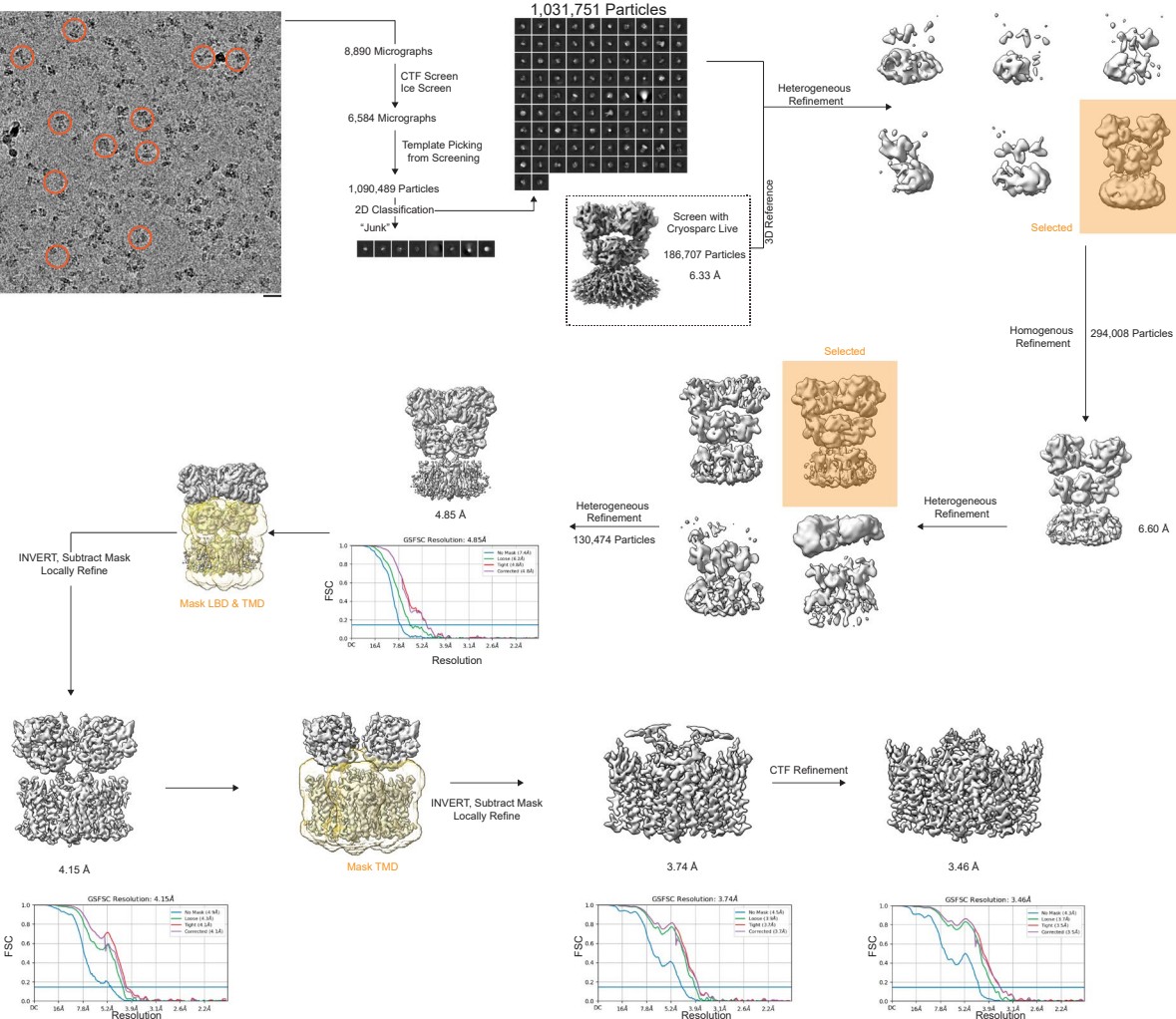

**Extended Data Fig. 4 | Cryo-EM processing workflow for GluA2-γ2$_{IS-2}$.** Image processing workflow and approach in Cryosparc.

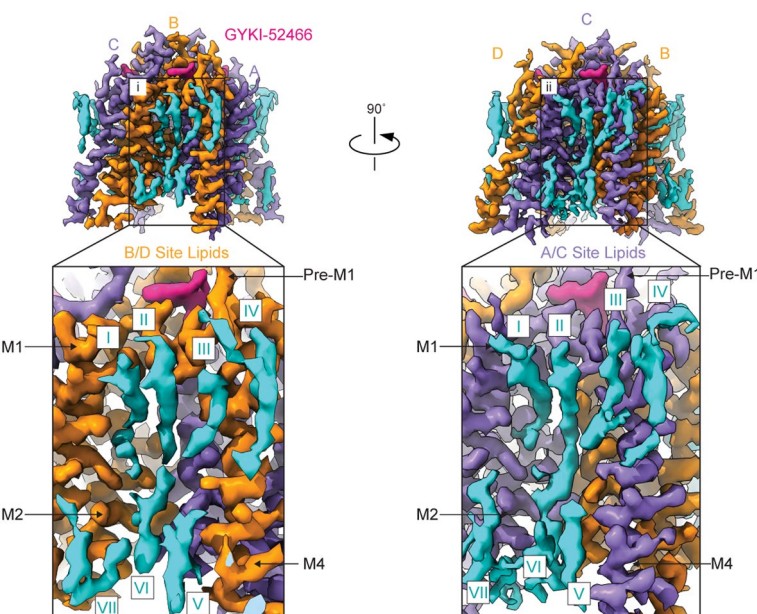

**Extended Data Fig. 5 | Structured lipids stabilize the AMPAR TMD.** Coulomb potential maps of the AMPAR TMD from signal subtraction and focused refinement highlighting the presence of lipids (blue) bound to the TMD. (i) close-up of lipids bound to the B/D TMD subunits of the receptor showing at least seven distinct densities. (ii) close-up of the A/C subunits of the receptor demonstrating similar, but distinct lipid arrangements around the TMD.

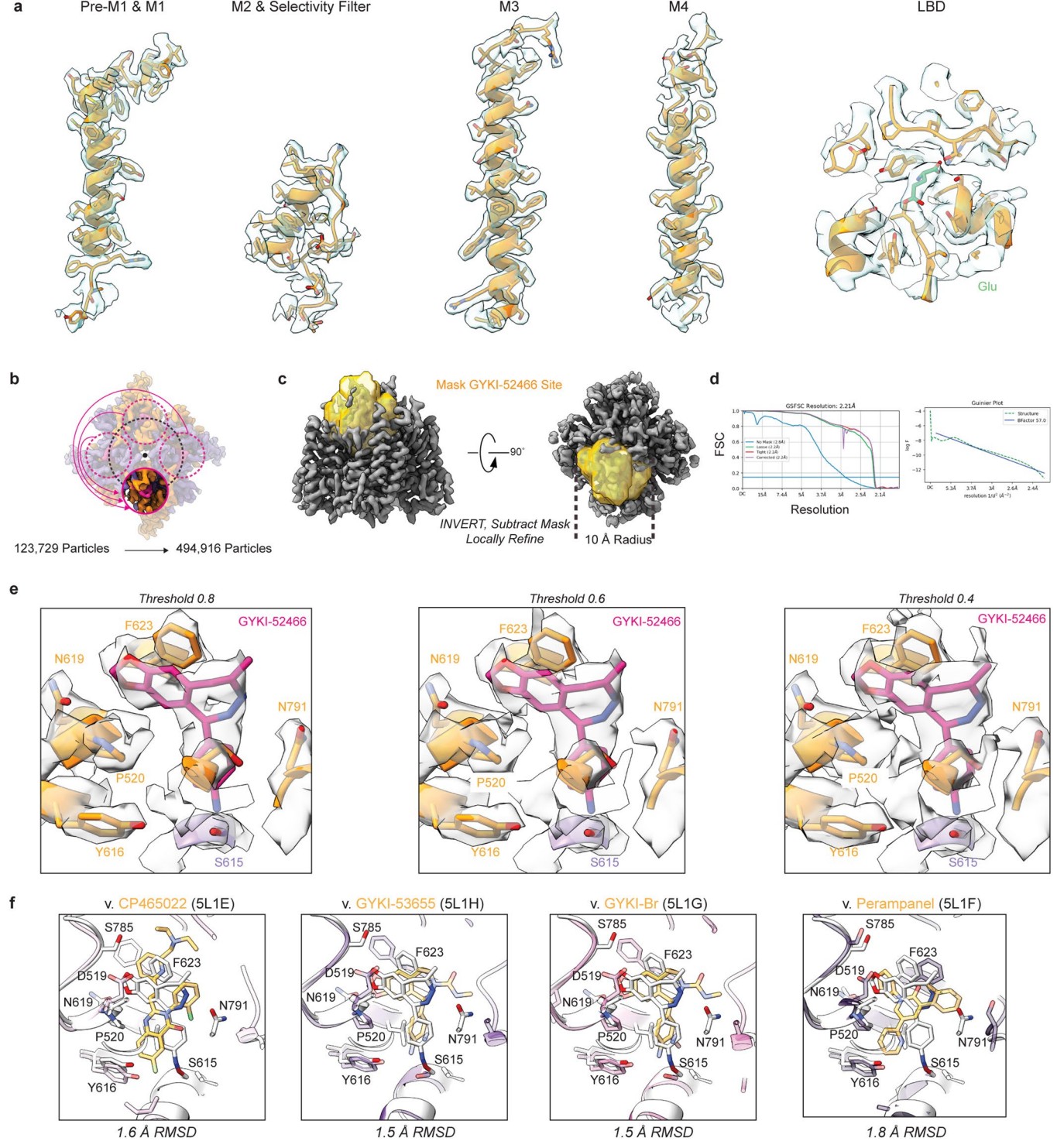

**Extended Data Fig. 6 | GluA2-γ2$_{IS-1}$ map examples and workflow for elucidating GYKI-52466 binding pocket.** (**a**) Examples of Cryo-EM map for the AMPAR TMD and Glu-bound LBD. (**b**) Symmetry expansion was applied through to the isolated GluA2-γ2$_{IS-1}$ TMD to increase the effective particle count of the GYKI-52466 binding pocket. (**c**) Following expansion one of the four GYKI-52466 binding pockets was masked and then the mask inverted to subtract away the remaining TMD structure. (**d**) Local refinement of the isolated GYKI-52466 binding pocket resolved the pocket to 2.21 Å resolution. (**e**) Cryo-EM map of the GYKI-52466 pocket shown from left to right at thresholds of 0.8, 0.6, and 0.4. (**f**) Comparisons between PPLM binding pockets and GYKI-52466 from this study.

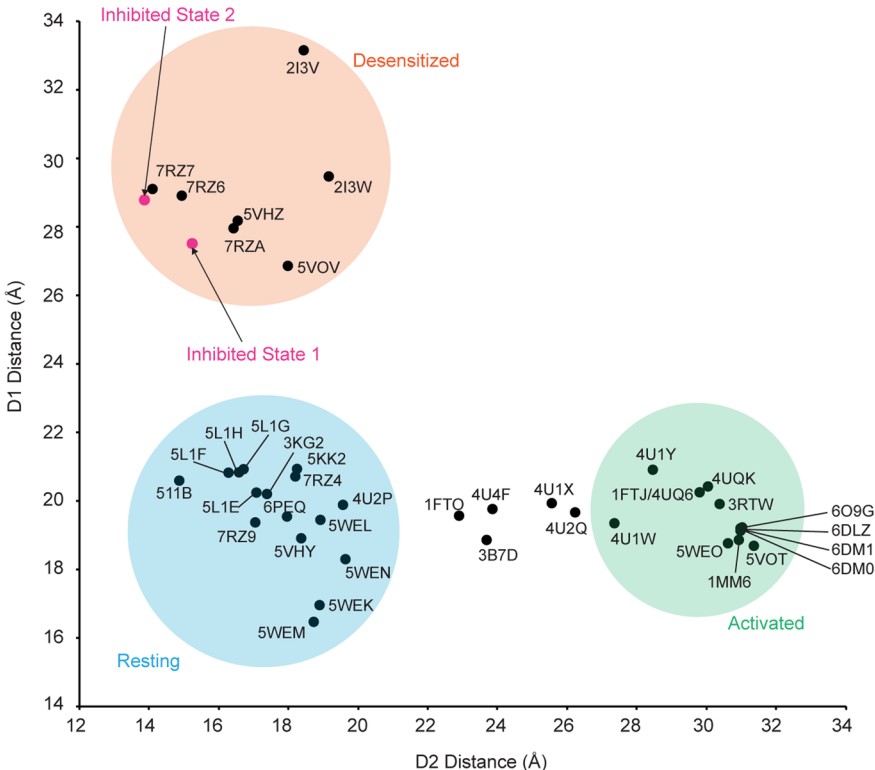

**Extended Data Fig. 7 | Clustering of AMPAR structures based on D1 and D2 distances.** A detailed look of how published AMPAR structures cluster based on the measurements between the D1 and D2 lobes of LBD clamshells within local dimers. PDB reference numbers are given for each structure measured.

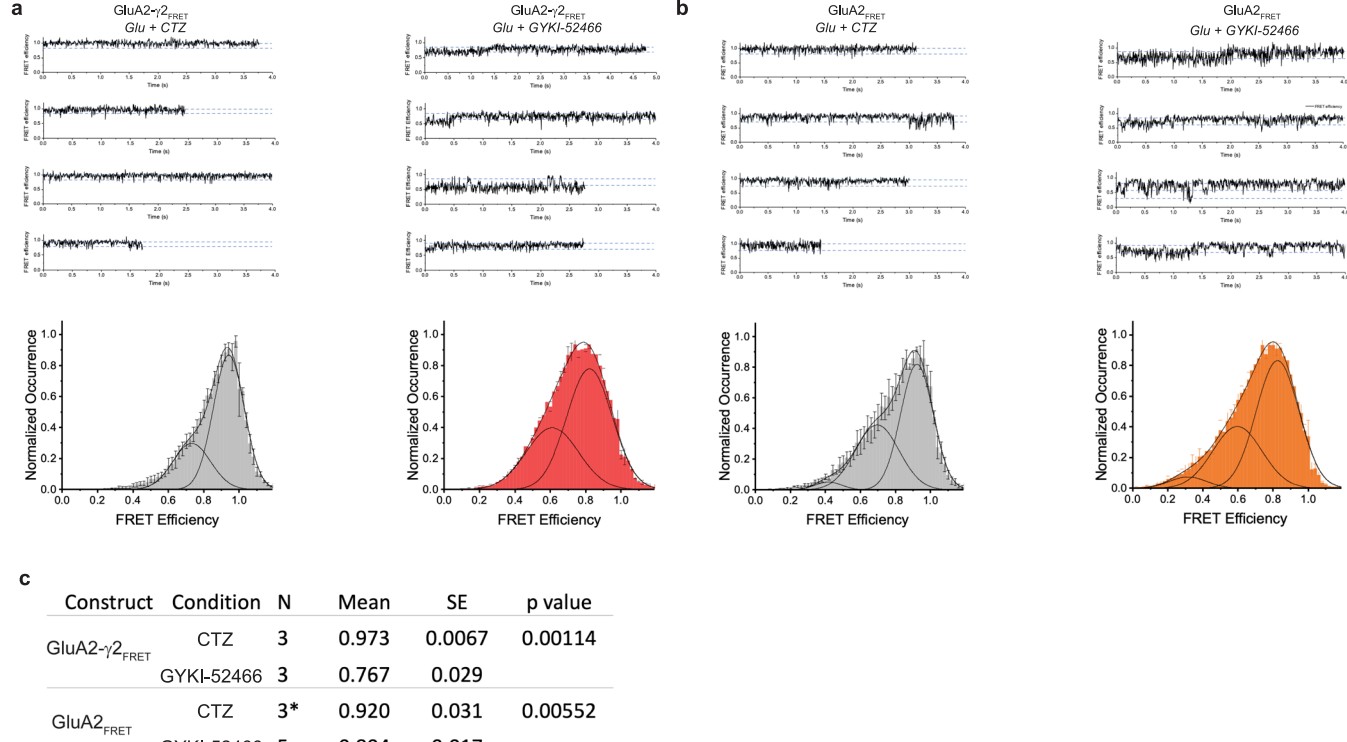

| Construct | Condition | N | Mean | SE | p value |
|---|---|---|---|---|---|
| GluA2-γ2$_{FRET}$ | CTZ | 3 | 0.973 | 0.0067 | 0.00114 |
| | GYKI-52466 | 3 | 0.767 | 0.029 | |
| GluA2$_{FRET}$ | CTZ | 3* | 0.920 | 0.031 | 0.00552 |
| | GYKI-52466 | 5 | 0.804 | 0.017 | |

**Extended Data Fig. 8 | smFRET characterization of GYKI-52466 and CTZ allosteric modulation of AMPARs. (a)** (Top) Representative FRET efficiency traces for sampled four molecules in each condition (1 mM Glu + 100 μM CTZ or 100 μM GYKI-52466) in GluA2-γ2$_{FRET}$. (Bottom) FRET efficiency histogram generated from the compilation of all analyzed single-molecule traces with FRET efficiency traces obtained using MDL from STaSI[93]. Data are represented as mean values +/− SEM across multiple days. **(b)** Same as panel a, but GluA2$_{FRET}$. **(c)** Statistical analysis of the smFRET data. Mean of the mode for each day with

standard error demonstrates a significant decrease in FRET efficiency from CTZ condition to GYKI-52466 condition using two-sample t-test assuming a one-tail distribution with known variances. For GluA2-γ2$_{FRET}$ $t = 6.931, df = 4, p = 0.00114$, for GluA2$_{FRET}$ $t = 3.625, df = 6, p = 0.00552$. The number of molecules included in the analysis for each condition is as follows: GluA2-γ2$_{FRET}$ (CTZ = 76, GYKI-52466 = 77), GluA2$_{FRET}$ (CTZ = 62*, GYKI-52466 = 96). *30 molecules with 1 mM of glutamate and 100 μM CTZ were obtained from Carrillo and Shaikh et al.[60].

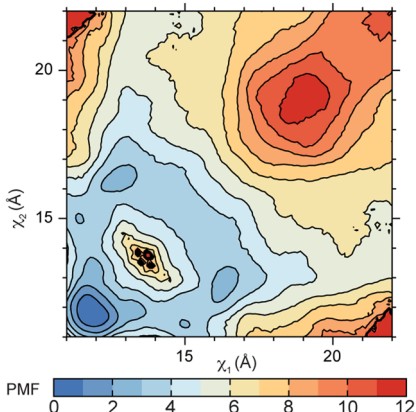

**Extended Data Fig. 9 | Free energy landscape governing desensitization in the GluA2-L483Y LBD dimer.** The PMF is computed as a function of $(c_1, c_2)$, the two distances between helices D and J at the dimer interface. The PMF is contoured in 1 kcal/mol increments.

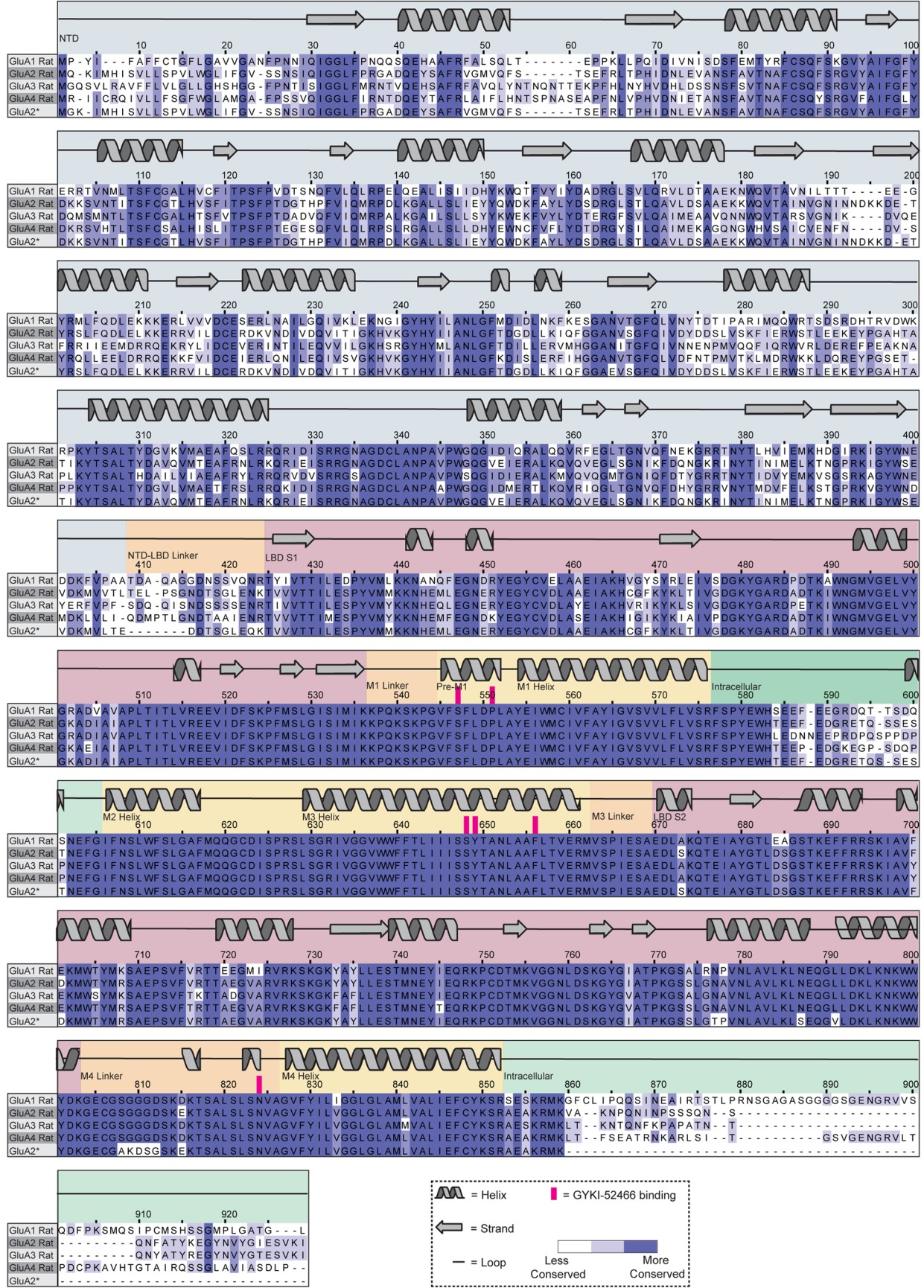

**Extended Data Fig. 10 | Alignment of GluA subunits.** Multiple sequence alignment of Rat GluA1-4 protein sequences. Conservation is indicated by the intensity of purple coloring. Secondary structure is displayed above the alignment. GYKI-52466 interacting residues are highlighted in pink.

# Reporting Summary

## Statistics

For all statistical analyses, confirm that the following items are present in the figure legend, table legend, main text, or Methods section.

| n/a | Confirmed | |
|---|---|---|
| ☐ | ☒ | The exact sample size (*n*) for each experimental group/condition, given as a discrete number and unit of measurement |
| ☐ | ☒ | A statement on whether measurements were taken from distinct samples or whether the same sample was measured repeatedly |
| ☐ | ☒ | The statistical test(s) used AND whether they are one- or two-sided *Only common tests should be described solely by name; describe more complex techniques in the Methods section.* |
| ☒ | ☐ | A description of all covariates tested |
| ☐ | ☒ | A description of any assumptions or corrections, such as tests of normality and adjustment for multiple comparisons |
| ☐ | ☒ | A full description of the statistical parameters including central tendency (e.g. means) or other basic estimates (e.g. regression coefficient) AND variation (e.g. standard deviation) or associated estimates of uncertainty (e.g. confidence intervals) |
| ☐ | ☒ | For null hypothesis testing, the test statistic (e.g. *F*, *t*, *r*) with confidence intervals, effect sizes, degrees of freedom and *P* value noted *Give P values as exact values whenever suitable.* |
| ☒ | ☐ | For Bayesian analysis, information on the choice of priors and Markov chain Monte Carlo settings |
| ☒ | ☐ | For hierarchical and complex designs, identification of the appropriate level for tests and full reporting of outcomes |
| ☒ | ☐ | Estimates of effect sizes (e.g. Cohen's *d*, Pearson's *r*), indicating how they were calculated |

*Our web collection on statistics for biologists contains articles on many of the points above.*

## Software and code

Policy information about availability of computer code

| Data collection | EPU-3.5, CHARMM36, pCLAMP10 |
|---|---|
| Data analysis | Cryosparc-4.2.1, ChimeraX-1.5, Isolde-1.6, COOT-0.9.8.2, Phenix-1.20, Biopython.pdb, Jalview-2.11.3, Molprobity-4.5.2, OriginPro 2023b, CLampfit-11, SCWRL4, ModLoop Server, CHARMM, T-Coffee, MOLE Online |

For manuscripts utilizing custom algorithms or software that are central to the research but not yet described in published literature, software must be made available to editors and reviewers. We strongly encourage code deposition in a community repository (e.g. GitHub). See the Nature Portfolio guidelines for submitting code & software for further information.

## Data

Policy information about availability of data

All manuscripts must include a data availability statement. This statement should provide the following information, where applicable:
- Accession codes, unique identifiers, or web links for publicly available datasets
- A description of any restrictions on data availability
- For clinical datasets or third party data, please ensure that the statement adheres to our policy

All cryo-EM reconstructions are deposited into the Electron Microscopy Data Bank (EMDB) and will be released upon publication. The accession codes for GluA2-γ2IS-1 and GluA2-γ2IS-2 are EMDB-43275 and EMDB-43276, respectively. The full maps (prior to local refinements and signal subtraction) are the primary cryo-EM maps in each deposition and each local map are supplied as supplemental files in each deposition. The GluA2-γ2IS-1 and GluA2-γ2IS-2 are deposited in the Protein

Data Bank and will be released upon publication. The PDB access codes for GluA2-γ2IS-1 and GluA2-γ2IS-2 are 8VJ6 and 8VJ7, respectively. All conformers from MD simulation trajectories, data from umbrella sampling, analysis code, will be publicly available from Zenodo upon publication of this work accession #10967297. DOI: 10.5281/zenodo.1096729

# Research involving human participants, their data, or biological material

Policy information about studies with <u>human participants or human data</u>. See also policy information about <u>sex, gender (identity/presentation), and sexual orientation</u> and <u>race, ethnicity and racism</u>.

| | |
|---|---|
| Reporting on sex and gender | N/A |
| Reporting on race, ethnicity, or other socially relevant groupings | N/A |
| Population characteristics | N/A |
| Recruitment | N/A |
| Ethics oversight | N/A |

Note that full information on the approval of the study protocol must also be provided in the manuscript.

# Field-specific reporting

Please select the one below that is the best fit for your research. If you are not sure, read the appropriate sections before making your selection.

☒ Life sciences  ☐ Behavioural & social sciences  ☐ Ecological, evolutionary & environmental sciences

For a reference copy of the document with all sections, see <u>nature.com/documents/nr-reporting-summary-flat.pdf</u>

# Life sciences study design

All studies must disclose on these points even when the disclosure is negative.

| | |
|---|---|
| Sample size | Sample size was not predetermined prior to study, but was determined by the availability of microscope time. For smFRET, sample size was dictated by molecules with a single photobleaching event. This stringent criterion ensured that only one donor and one acceptor fluorophore were attached to each GluA2 protein. For electrophysiology of our cryo-EM construct, we predetermined to record each condition in triplicate in lieu of statistic-based sample size determination. Samples were patched if they fluoresced green from the cryo-EM construct. |
| Data exclusions | No data was excluded. |
| Replication | Image processing in cryo-EM was duplicated and performed with ab initio models generated from the data. No external data was input into the image processing. All successful electrophysiological recordings were reproducible. Dose response recordings for allosteric competition were repeated in triplicate. Sweeps per cell are also reported. For smFRET, replaces were: GluA2-γ2FRET (CTZ = 76, GYKI-52466 = 77), GluA2FRET (CTZ = 62*, GYKI-52466 = 96). 30 molecules with 1 mM of glutamate and 100 μM CTZ were obtained from Carrillo and Shaikh et al (2020). All attempts to reproduce experimental data were successful. |
| Randomization | These experiments were not randomized. Covariates were minimized by comparing different conditions on the same experimental day to minimize batch effects. |
| Blinding | The investigators were not blinded to the data analysis. This is not technically or practically feasible for Cryo-EM, patch clamp, smFRET, or Molecular Dynamics studies. Researchers conducting the data analysis for each experiment were also responsible for data collection, making blinding impossible. |

# Reporting for specific materials, systems and methods

We require information from authors about some types of materials, experimental systems and methods used in many studies. Here, indicate whether each material, system or method listed is relevant to your study. If you are not sure if a list item applies to your research, read the appropriate section before selecting a response.

## Materials & experimental systems

| n/a | Involved in the study |
|---|---|
| ☐ | ☒ Antibodies |
| ☐ | ☒ Eukaryotic cell lines |
| ☒ | ☐ Palaeontology and archaeology |
| ☒ | ☐ Animals and other organisms |
| ☒ | ☐ Clinical data |
| ☒ | ☐ Dual use research of concern |
| ☒ | ☐ Plants |

## Methods

| n/a | Involved in the study |
|---|---|
| ☒ | ☐ ChIP-seq |
| ☒ | ☐ Flow cytometry |
| ☒ | ☐ MRI-based neuroimaging |

## Antibodies

| Antibodies used | biotinylated Goat Anti-Mouse IgG (H + L) secondary antibody (Jackson Immunoresearch Laboratories, catalog number 115-065-003) used at 2.7ng/ul<br>anti-GluR2 , Clone: L21/32(BioLegend®) - used at 3.0ng/ul<br>anti-TARPγ2, Clone: N245/36 (Millipore) - used at 2.4ng/ul |
|---|---|
| Validation | biotinylated Goat Anti-Mouse IgG (H + L) secondary antibody - From manufacturer: Based on immunoelectrophoresis and/or ELISA, the antibody reacts with whole molecule mouse IgG. It also reacts with the light chains of other mouse immunoglobulins. No antibody was detected against non-immunoglobulin serum proteins. The antibody may cross-react with immunoglobulins from other species.<br><br>anti-GluR2 - From manufacturer: Cross-reacts with Human, Mouse, Rat; Each lot of this antibody is quality control tested by Western blotting; It does not cross-react with GluA1/GluR1, GluA3/GluR3, or GluA4/GluR4.; References: 1. Brown EA, et al. 2018. Mol Autism. 9:48. 2. Lautz JD, et al. 2021. Cell Rep. 37:110076.<br><br>anti-TARPγ2- From manufacturer:Each new lot of antibody is quality control tested by western blot on rat whole brain lysate and confirmed to stain the expected molecular weight band. |

## Eukaryotic cell lines

Policy information about cell lines and Sex and Gender in Research

| Cell line source(s) | Sf9 for baculorvirus (Gibco, A35243); Expi293 Gnti- for protein overexpression (Gibco, A39240). HEK293T (ATCC, CRL-3216) for smFRET and electrophysiology. |
|---|---|
| Authentication | Sf9s cells are routinely used in our labs and were not specifically validated for these studies outside of the manufacturer's specifications. Expi293 Gnti- cells are routinely used in our labs and were not specifically validated for these studies outside of the manufacturer's specifications. |
| Mycoplasma contamination | Sf9 cell lines tested negative for mycoplasma. Expi293 Gnti- cell lines tested negative for mycoplasma. HEK293T tested negative for mycoplasma. |
| Commonly misidentified lines<br>(See ICLAC register) | No commonly misidentified lines were used for this study. |

## Plants

| Seed stocks | N/A |
|---|---|
| Novel plant genotypes | N/A |
| Authentication | N/A |

