## [Peer Review File · Nature Structural & Molecular Biology]

Peer Review Information

Manuscript Title: Allosteric Competition and Inhibition in AMPA Receptors

Corresponding author name(s): Edward Twomey, Richard Huganir, Albert Lau

Reviewer Comments & Decisions:

Decision Letter, initial version:

Message: 14th Feb 2024

Dear Dr. Twomey,

Thank you again for submitting your manuscript "Allosteric Competition and Inhibition in AMPA Receptors". We now have comments (below) from the 2 reviewers who evaluated your paper. In light of those reports, we remain interested in your study and would like to see your response to the comments of the referees, in the form of a revised manuscript.

You will see that while reviewers appreciate the results, they raise several concerns which will need to be addressed in a revision. Specifically, please take into account suggestions of reviewer #1 to clarify several aspects of the manuscript, and acknowledge previous work in the field. Please revisit electrophysiology data on the effects of GYKI and CTZ, in line with reviewer #2 comments.

Please be sure to address/respond to all concerns of the referees in full in a point-by-point response and highlight all changes in the revised manuscript text file. If you have comments that are intended for editors only, please include those in a separate cover letter.

We expect to see your revised manuscript within 12 weeks. If you cannot send it within this time, please contact us to discuss an extension; we would still consider your revision, provided that no similar work has been accepted for publication at NSMB or published elsewhere.

Reporting Summary:

When submitting the revised version of your manuscript, please pay close attention to our [Digital Image Integrity Guidelines](https://www.nature.com/nature-portfolio/editorial-policies/image-integrity). and to the following points below:

Please note that all key data shown in the main figures as cropped gels or blots should be presented in uncropped form, with molecular weight markers. These data can be aggregated into a single supplementary figure item. While these data can be displayed in a relatively informal style, they must refer back to the relevant figures. These data should be submitted with the final revision, as source data, prior to acceptance, but you may want to start putting it together at this point.

Data availability: this journal strongly supports public availability of data. All data used in accepted papers should be available via a public data repository, or alternatively, as Supplementary Information. If data can only be shared on request, please explain why in your Data Availability Statement, and also in the correspondence with your editor. Please note that for some data types, deposition in a public repository is mandatory - more information on our data deposition policies and available repositories can be found below: <https://www.nature.com/nature-research/editorial-policies/reporting-standards#availability-of-data>

Nature Structural & Molecular Biology is committed to improving transparency in authorship. As part of our efforts in this direction, we are now requesting that all authors identified as 'corresponding author' on published papers create and link their Open Researcher and Contributor Identifier (ORCID) with their account on the Manuscript Tracking System (MTS), prior to acceptance. This applies to primary research papers only. ORCID helps the scientific community achieve unambiguous attribution of all scholarly contributions. You can create and link your ORCID from the home page of the MTS by clicking on 'Modify my Springer Nature account'. For more information please visit please visit www.springernature.com/orcid.

[redacted]

Sincerely,

Katarzyna Ciazynska, PhD
(she/her)
Associate Editor
Nature Structural & Molecular Biology
<https://orcid.org/0000-0002-9899-2428>

Referee expertise:

Referee #1: AMPARs, structural biology

Referee #2: AMPARs, neurobiology

Reviewers' Comments:

Reviewer #1:

Remarks to the Author:

In this study the authors report the structure of an AMPAR bound by glutamate and a non-competitive inhibitor PPLM called GYKI-52466. Previously, structures have only been reported for PPLMs bound to the AMPAR in the absence of glutamate. From the previous studies the pocket locations and binding modes at the top of the TMD are known for PPLMs, as well as a proposed mechanism of non-competitive inhibition by blocking communication from the ATD to the TMD. This new structure that includes glutamate provides new insight on the mechanism of non-competitive inhibition by PPLMs because the LBD layer adopts a conformation that is distinct from either the apo resting, activated or desensitised conformation. The LBD dimers represent the desensitised state, as is normally seen for AMPARs bound by glutamate (Fig. 3b-d). However, the LBD layer as a whole (i.e. relative orientation of the two dimers) is different from the desensitised state (Fig. 3e). This also explains why the positive allosteric modulator, CTZ, is absent from the structure even though it was included in the cryo-EM condition, because this conformation disrupts the CTZ binding pocket. FRET functional analysis is used to further validate the disruption to the CTZ pocket.

The TMDs are a bit different in this structure versus those for TMDs in resting state structures bound by PPLMs (EDF 2 – RMSD ~ 2Å). The TMDs are similar to open TMDs except for at the open channel gate (Fig. 3a). However, the authors do not compare the TMDs to those in desensitised receptors, which would be helpful for considering the conformation.

The cryo-EM map data allows for good confidence in the analysis and interpretation of the models. The figures and analysis are generally clear.

Overall, it seems that non-competitive inhibition induces a new conformation of the LBD layer, which prevents both activation and desensitisation of the LBD layer, resulting in a new LBD layer state. This message could be more clearly provided in the paper in general (in line with it being shown as such in Fig.5 and 6). It is also unclear if this process also induces a new conformation of the TMD because there is no comparison to the desensitized TMD and it is not clearly discussed whether the TMD is a distinct state from those previously solved.

The paper could play to its strengths more whilst also acknowledging what has already been established in the field. For example, it is mentioned on p.9, line 23, that "the mechanisms of allosteric inhibition by negative allosteric modulators have remained

unknown". This is unfair on the previous literature and structures showing some insight to this process, such as that PPLMs wedge at the top of the TMD and block communication down from the LBD. However, what has not been looked at, and is a unique question addressed here, is the structural allosteric interplay between agonist, PAM and non-competitive inhibition. Another example is in the introduction, p.2, line 28 "While the binding sites of PPLMs have been generally described the precise mechanism by which PPLMs inhibit AMPAR function is unknown." Actually, the previous literature does provide insight into the mechanism (wedge block of LBD communication with the TMD), but what it doesn't provide, and what is uniquely provided in this study is the allosteric interplay between agonist, PAM and non-competitive inhibition. It would be good to clarify the main message of the paper throughout.

As a general point, the figures, legends and main text were all in different places in the (all-in-one) document, and the figures had no figure numbers on them. This made it very hard to scroll between the various sections on the computer and evaluate the manuscript.

To summarize, AMPA-Rs are of major importance within the ion channel structure field and this study provides novel insight into a mechanism of non-competitive inhibition that should be of interest to many, given that the structure represents a distinct state. The structural data seem clear-cut and are supported by functional data. In my opinion this study meets the criteria for publication in NSMB, but the manuscript needs a little bit of work in clarifying its message, as discussed above and also addressed in the points below.

Some specific points are:

- Abstract:

o Meaning of "desensitises AMPARs to activation" is unclear

o I would prefer it if the abstract captured the main message of the paper more clearly, i.e. ...induced a state in the LBD layer distinct from previous resting, open, desensitized conformations...(because this message is not coming through as clearly as it could in the paper).

- P3, line 45 – the claim this provides a framework for studying inhibition of "and other ligand-gated ion channels" is unfounded and should be removed.

- P4, line 17-20 – unclear

- P4, line 21 – mention curve fit approach in methods instead.

- P4, line 21-25 – The authors cite literature showing the CTZ shifts the GYKI-52466 IC50 10-fold for wild-type constructs, but they do not show this for their cryo-EM construct.

Given that the premise of the paper is that their cryo-EM construct structure shows an allosteric competition between these two compounds it would be good to confirm it functionally in the cryo-EM construct, i.e. obtain one more DR for Glu+GYKI-52466 without CTZ, which should in theory be 10-fold shifted (unless there is a reason to not do this, such as it is particularly challenging when factoring in the glutamate desensitisation in the absence of CTZ).

- P5, lines 22-30 versus Fig.2a:

o What is the distance between F623 and the molecule?

o Molecule would be clearer without the spheres/space-fill.

o P520 is hidden but should be shown.

- Regarding the binding mode, a clearer contextualisation would be helpful:

o Have structures of GYKI-52466 been shown specifically previously (or were they of other different PPLMs)?

o Does GYKI-52466 have any differences in affinity vs other PPLMs and is there any

differences in potential contacts in the pocket versus other PPLMs that could hint at why this is? This should be mentioned.

o Is the conformation of the pocket different or the same from previously published pockets in resting state receptors (without Glu bound) and does this change the contacts? Figure with overlay and RMSD of the pocket would be helpful.

- P.6, line 8: It is not mentioned what the activated state is bound by in order to activate it (Glu + CTZ?) in the main text or the legend for 3a. This should be stated.

- P.6, line 10: "largely similar" -> State TMD RMSD.

- P.6, line 16: "similar" -> State LBD RMSD.

- Fig. 3e: I cannot figure out what is happening. Activated is written in grey. Desensitized is written in blue. But there are orange and purple subunits most clearly visible. And then pink arrows for inhibition. Split into two separate panels?

- P.9, line 40: Is it a "desensitized-like" state or its own distinct "non-competitively inhibited" state. The results suggest it's a distinct state (Fig. 5a, inhibited state 1). Better to avoid calling it a "desensitized-like" state which might become mis-quoted in the literature, e.g. "shunts it into a distinct allosterically inhibited state", etc.

Dr Paul Steven Miller

Reviewer #2:

Remarks to the Author:

The manuscript by Hale and colleagues uses a combination of cryo-EM, FRET, electrophysiology, and molecular dynamics simulations to elucidate the mechanisms by which GYKI-52466 (a perampanel-like molecule) exerts negative allosteric modulation of AMPA receptors through binding in the transmembrane domain and forcing the LBD clamshells into a desensitized-like state upon glutamate binding. Further, the data demonstrates of rearrangements in the LBD due to GYKI-52466 binding prevents positive allosteric modulation by cyclothiazide by disrupting the cyclothiazide binding site. The study is high quality, well-written and illustrated, and in my opinion, will be of very high interest to the glutamate receptor field. The data also lay the framework for future drug design. There are a few concerns about presentation and interpretation.

Major

1. In the results, it is written that "the effect of CTZ on desensitization... is negated by GYKI-52466". Later in the discussion (p. 10, lines 29-30). "GYKI-52466 prevents CTZ from positively modulating AMPARs through rupturing the CTZ binding site". But this is not illustrated in Extended data figure 1. GYKI appears to reduce the AMPAR current generated in response to application of glutamate + CTZ, but the effect of CTZ in preventing (or slowing) desensitization is still present in GYKI. In these recordings, is the slowing of desensitization due to GYKI alone if CTZ is no longer bound? Or is the interpretation that GYKI is negating CTZ's effect too oversimplified?

2. In extended data figure 1b, it appears the bar to indicate drug application is shifted to the right. It also appears that GYKI in the presence of CTZ prolonged the current after removal of glutamate. Is this observed in grouped data? If this is the case, it is important to note for readers since producing a persistent (albeit small) current could have dramatic effects on subthreshold depolarization.

Minor

1. It may be of the interest to the future of the field to have a more descriptive or specific

word for GYKI's desensitized-like state than "inhibition". The data describe a new conformational state of the receptor that should be named.

2. It would be interesting to include speculation in the discussion about the effect of CNQX or DNQX with GYKI. Would the partial cleft closure from these "antagonists" be sufficient to destabilize the D1-D1 dimer interface?

Typos

1. Page 2, lines 25-26. Perampanel treatment produces unwanted side effects but from my understanding for many patients with GRI disorders (and their families and caregivers), it has been very beneficial.

2. "dose-response" should be "concentration-response"

3. Page 5, the small paragraph that starts on line 50 could be moved up to line 10 on the same page for ease of readership.

4. Page 8, line 5, "activate"

Author Rebuttal to Initial comments

A point-by-point description (in blue) of how reviewer feedback was addressed follows:

Reviewer #1:

Remarks to the Author:

In this study the authors report the structure of an AMPAR bound by glutamate and a non-competitive inhibitor PPLM called GYKI-52466. Previously, structures have only been reported for PPLMs bound to the AMPAR in the absence of glutamate. From the previous studies the pocket locations and binding modes at the top of the TMD are known for PPLMs, as well as a proposed mechanism of non-competitive inhibition by blocking communication from the ATD to the TMD. This new structure that includes glutamate provides new insight on the mechanism of non-competitive inhibition by PPLMs because the LBD layer adopts a conformation that is distinct from either the apo resting, activated or desensitised conformation. The LBD dimers represent the desensitised state, as is normally seen for AMPARs bound by glutamate (Fig. 3b-d). However, the LBD layer as a whole (i.e. relative orientation of the two dimers) is different from the desensitised state (Fig. 3e). This also explains why the positive allosteric modulator, CTZ, is absent from the structure even though it was included in the cryo-EM condition, because this conformation disrupts the CTZ binding pocket. FRET functional analysis is used to further validate the disruption to the CTZ pocket.

The TMDs are a bit different in this structure versus those for TMDs in resting state structures bound by PPLMs (EDF 2 – RMSD ~ 2Å). The TMDs are similar to open TMDs except for at the open channel gate (Fig. 3a). However, the authors do not compare the TMDs to those in desensitised receptors, which would be helpful for considering the conformation.

We have now added panels to Extended Data 2 showing a comparison of the inhibited and desensitized TMDs. We have also added a discussion of this comparison to the results (p.7 lines 4-7).

The cryo-EM map data allows for good confidence in the analysis and interpretation of the models. The figures and analysis are generally clear.

Overall, it seems that non-competitive inhibition induces a new conformation of the LBD layer, which prevents both activation and desensitisation of the LBD layer, resulting in a new LBD layer state. This message could be more clearly provided in the paper in general (in line with it being shown as such in Fig.5 and 6). It is also unclear if this process also induces a new conformation of the TMD because there is no comparison to the desensitized TMD and it is not clearly discussed whether the TMD is a distinct state from those previously solved.

We added comparisons to the desensitized state (discussed above), and make it clear in the text that the overall state of the TMD is not distinct.

The paper could play to its strengths more whilst also acknowledging what has already been established in the field. For example, it is mentioned on p.9, line 23, that “the mechanisms of allosteric inhibition by negative allosteric modulators have remained unknown”. This is unfair on the previous literature and structures showing some insight to this process, such as that PPLMs wedge at the top of the TMD and block communication down from the LBD. However, what has not been looked at, and is a unique question addressed here, is the structural allosteric interplay between agonist, PAM and non-competitive inhibition. Another example is in the introduction, p.2, line 28 “While the binding sites of PPLMs have been generally described the precise mechanism by which PPLMs inhibit AMPAR function is unknown.” Actually, the previous literature does provide insight into the mechanism (wedge block of LBD communication with the TMD), but what it doesn’t provide, and what is uniquely provided in this study is the allosteric interplay between agonist, PAM and non-competitive inhibition. It would be good to clarify the main message of the paper throughout.

This is excellent feedback for us, and we made the changes throughout the abstract, introduction, text, and discussion.

As a general point, the figures, legends and main text were all in different places in the (all-in-one) document, and the figures had no figure numbers on them. This made it very hard to scroll between the various sections on the computer and evaluate the manuscript.

We agree and apologize for this.

To summarize, AMPA-Rs are of major importance within the ion channel structure field and this study provides novel insight into a mechanism of non-competitive inhibition that should be of interest to many, given that the structure represents a distinct state. The structural data seem

clear-cut and are supported by functional data. In my opinion this study meets the criteria for publication in NSMB, but the manuscript needs a little bit of work in clarifying its message, as discussed above and also addressed in the points below.

We thank the reviewer for this constructive feedback.

Some specific points are:

- Abstract:

o Meaning of “desensitises AMPARs to activation” is unclear

We removed this in the revised abstract.

o I would prefer it if the abstract captured the main message of the paper more clearly, i.e. ...induced a state in the LBD layer distinct from previous resting, open, desensitized conformations...(because this message is not coming through as clearly as it could in the paper).

We have rewritten the abstract to address this point and hope it communicates the key findings of the paper more clearly.

- P3, line 45 – the claim this provides a framework for studying inhibition of “and other ligand-gated ion channels” is unfounded and should be removed.

This phrase is absent from the revised version.

- P4, line 17-20 – unclear

This paragraph (p4 lines 11-27) has been rewritten for improved clarity.

- P4, line 21 – mention curve fit approach in methods instead.

Mention of the curve fit has been removed from the results section and is treated in the methods section.

- P4, line 21-25 – The authors cite literature showing the CTZ shifts the GYKI-52466 IC50 10-fold for wild-type constructs, but they do not show this for their cryo-EM construct. Given that the premise of the paper is that their cryo-EM construct structure shows an allosteric competition between these two compounds it would be good to confirm it functionally in the cryo-EM construct, i.e. obtain one more DR for Glu+GYKI-52466 without CTZ, which should in theory be

10-fold shifted (unless there is a reason to not do this, such as it is particularly challenging when factoring in the glutamate desensitisation in the absence of CTZ).

While we do not directly show this with our cryo-EM construct, our measurements are directly in line with published electrophysiology studies on wild-type AMPAR-TARP complexes (e.g., Knopp et al 2019). In addition, studies on the GluA2 part of our cryo-EM construct (without TARPy2) show similar behavior in the context of perampanel and CTZ (Yelshanskaya et al 2016).

Beyond this literature, the concentration response curve we present (Glu+GYKI-52466+CTZ), in combination with the cryo-EM, smFRET, and electrophysiology on wild-type constructs all point to the allosteric competition mechanism.

- P5, lines 22-30 versus Fig.2a:
 - o What is the distance between F623 and the molecule?
 - o Molecule would be clearer without the spheres/space-fill.
 - o P520 is hidden but should be shown.

We have addressed all these points in a revised Fig 2a and 2b.

- Regarding the binding mode, a clearer contextualisation would be helpful:
 - o Have structures of GYKI-52466 been shown specifically previously (or were they of other different PPLMs)?

Previously published structures have so far only resolved other PPLMs. We have pointed this out in the main text (p5 line 37-38).

- o Does GYKI-52466 have any differences in affinity vs other PPLMs and is there any differences in potential contacts in the pocket versus other PPLMs that could hint at why this is? This should be mentioned.

Yes, GYKI-52466 is the weakest of the PPLMs in terms of IC50 and affinity, and this is likely due to reduced contacts in the binding site. We have addressed this in the text (p6, line 7-10).

- o Is the conformation of the pocket different or the same from previously published pockets in resting state receptors (without Glu bound) and does this change the contacts? Figure with overlay and RMSD of the pocket would be helpful.

The pockets are the same with minor but potentially significant differences. We have highlighted this in the text (p5 line 44-49). Additionally, we have added panels to Extended Data Fig 6f showing comparisons between the GYKI-52466 pocket in the inhibited state and the previously published PPLM structures with RMSDs labeled.

- P6, line 8: It is not mentioned what the activated state is bound by in order to activate it (Glu + CTZ?) in the main text or the legend for 3a. This should be stated.

We have now addressed this in both the main text (p6, line 29) and in the figure legend for Fig 3a.

- P.6, line 10: "largely similar" -> State TMD RMSD.

We have now added the RMSD value for this comparison to the main text (p6 line 27).

- P6, line 16: "similar" -> State LBD RMSD.

We have now added the RMSD value to the main text (p6 line 30).

- Fig. 3e: I cannot figure out what is happening. Activated is written in grey. Desensitised is written in blue. But there are orange and purple subunits most clearly visible. And then pink arrows for inhibition. Split into two separate panels?

We have revised Fig 3e for improved clarity by representing helices as cylinders and improving the color contrast.

- P.9, line 40: Is it a "desensitized-like" state or its own distinct "non-competitively inhibited" state. The results suggest it's a distinct state (Fig. 5a, inhibited state 1). Better to avoid calling it a "desensitized-like" state which might become mis-quoted in the literature, e.g. "shunts it into a distinct allosterically inhibited state", etc.

We thank the reviewer for this suggestion, which echoes a similar point made by Reviewer #2. We have decided to name this conformation the "allosterically inhibited" state to differentiate it from other states of the receptor.

Dr Paul Steven Miller

Reviewer #2:

Remarks to the Author:

The manuscript by Hale and colleagues uses a combination of cryo-EM, FRET, electrophysiology, and molecular dynamics simulations to elucidate the mechanisms by which GYKI-52466 (a perampanel-like molecule) exerts negative allosteric modulation of AMPA receptors through binding in the transmembrane domain and forcing the LBD clamshells into a desensitized-like state upon glutamate binding. Further, the data demonstrates of rearrangements in the LBD due to GYKI-52466 binding prevents positive allosteric modulation by cyclothiazide by disrupting the cyclothiazide binding site. The study is high quality, well-written and illustrated, and in my opinion, will be of very high interest to the glutamate receptor field. The data also lay the framework for future drug design. There are a few concerns about presentation and interpretation.

Major

1. In the results, it is written that “the effect of CTZ on desensitization... is negated by GYKI-52466”. Later in the discussion (p. 10, lines 29-30). “GYKI-52466 prevents CTZ from positively modulating AMPARs through rupturing the CTZ binding site”. But this is not illustrated in Extended data figure 1. GYKI appears to reduce the AMPAR current generated in response to application of glutamate + CTZ, but the effect of CTZ in preventing (or slowing) desensitization is still present in GYKI. In these recordings, is the slowing of desensitization due to GYKI alone if CTZ is no longer bound? Or is the interpretation that GYKI is negating CTZ’s effect too oversimplified?

We thank the reviewer for this suggestion and agree that this description was overly simplistic and have edited this paragraph (p4 lines 12-29) to better reflect the electrophysiology data.

As the reviewer suggests, there is also the possibility in our data that we cannot control for the off rates of both GYKI-52466 and CTZ. This could account for the heterogeneity that we observe in our cryo-EM data, but cannot reconstruct (pg 8, lines 34-37).

2. In extended data figure 1b, it appears the bar to indicate drug application is shifted to the right. It also appears that GYKI in the presence of CTZ prolonged the current after removal of glutamate. Is this observed in grouped data? If this is the case, it is important to note for readers since producing a persistent (albeit small) current could have dramatic effects on subthreshold depolarization.

After reviewing our electrophysiology data, we observed the prolonged current was not present in other measurements. Therefore, we have replaced the trace with a measurement that better represents the effect of 300 μ M GYKI and CTZ. Additionally, we performed two additional measurements for the 300 μ M GYKI +100 μ M CTZ + 1 mM glutamate condition to further verify this and these two measurements are added to the dose response curve and IC50 was calculated with these two additional measurements (the new value is within the standard deviation of what we reported previously).

Minor

1. It may be of the interest to the future of the field to have a more descriptive or specific word for GYKI's desensitized-like state than "inhibition". The data describe a new conformational state of the receptor that should be named.

We thank the reviewer for this suggestion which echoes a similar point made by Reviewer #1. To differentiate this state from competitive antagonist-bound states of the receptor we have decided to call our state the "allosterically inhibited" state.

2. It would be interesting to include speculation in the discussion about the effect of CNQX or DNQX with GYKI. Would the partial cleft closure from these "antagonists" be sufficient to destabilize the D1-D1 dimer interface?

This is an interesting idea by the reviewer. However, we do not expect significant destabilization to occur in this case. For example, in Gangwar et al 2023 a structure of an AMPAR was solved in complex with ZK/MPQX and perampanel, and the D1-D1 dimer interface is mostly intact.

Typos

1. Page 2, lines 25-26. Perampanel treatment produces unwanted side effects but from my understanding for many patients with GRI disorders (and their families and caregivers), it has been very beneficial.

We thank the reviewer for pointing this out and in response to this feedback, we have highlighted several studies in the main text that have successfully employed perampanel as a therapeutic for patients with genetic disorders including GRI and SynGAP1.

2. "dose-response" should be "concentration-response"

We have incorporated this suggestion into the revised manuscript.

3. Page 5, the small paragraph that starts on line 50 could be moved up to line 10 on the same page for ease of readership.

We have incorporated this suggestion into the revised manuscript.

4. Page 8, line 5, "activate"

We have incorporated this suggestion into the revised manuscript.

Decision Letter, first revision:

Message: Our ref: NSMB-A48691A

14th Mar 2024

Dear Dr. Twomey,

Thank you for submitting your revised manuscript "Allosteric Competition and Inhibition in AMPA Receptors" (NSMB-A48691A). It has now been seen by the original referees and their comments are below. The reviewers find that the paper has improved in revision, and therefore we'll be happy in principle to publish it in Nature Structural & Molecular Biology, pending minor revisions to satisfy the referees' final requests and to comply with our editorial and formatting guidelines.

Sincerely,

Katarzyna Ciazynska, PhD
(she/her)
Associate Editor
Nature Structural & Molecular Biology
<https://orcid.org/0000-0002-9899-2428>

Reviewer #1 (Remarks to the Author):

I am satisfied with the author rebuttal and improvements to the manuscript.

Final Decision Letter:

Message: 3rd May 2024

Dear Dr. Twomey,

We are now happy to accept your revised paper "Allosteric Competition and Inhibition in AMPA Receptors" for publication as an Article in Nature Structural & Molecular Biology.

Your paper will be published online soon after we receive proof corrections and will appear in print in the next available issue. You can find out your date of online publication by contacting the production team shortly after sending your proof corrections.

You may wish to make your media relations office aware of your accepted publication, in case they consider it appropriate to organize some internal or external publicity. Once your paper has been scheduled you will receive an email confirming the publication details. This is normally 3-4 working days in advance of publication. If you need additional notice of the date and time of publication, please let the production team know when you receive the proof of your article to ensure there is sufficient time to coordinate. Further

information on our embargo policies can be found here:
<https://www.nature.com/authors/policies/embargo.html>

Please note that *Nature Structural & Molecular Biology* is a Transformative Journal (TJ). Authors may publish their research with us through the traditional subscription access route or make their paper immediately open access through payment of an article-processing charge (APC). Authors will not be required to make a final decision about access to their article until it has been accepted. Find out more about Transformative Journals

Sincerely,

Katarzyna Ciazynska, PhD
(she/her)
Associate Editor
Nature Structural & Molecular Biology
<https://orcid.org/0000-0002-9899-2428>